# TIME SERIES CONTINUOUS MODELING FOR IMPUTATION AND FORECASTING WITH IMPLICIT NEURAL REPRESENTATIONS

## ABSTRACT

We introduce a novel modeling approach for time series imputation and forecasting, tailored to address the challenges often encountered in real-world data, such as irregular samples, missing data, or unaligned measurements from multiple sensors. Our method relies on a continuous-time-dependent model of the series' evolution dynamics. It leverages adaptations of conditional, implicit neural representations for sequential data. A modulation mechanism, driven by a meta-learning algorithm, allows adaptation to unseen samples and extrapolation beyond observed time-windows for long-term predictions. The model provides a highly flexible and unified framework for imputation and forecasting tasks across a wide range of challenging scenarios. It achieves state-of-the-art performance on classical benchmarks and outperforms alternative time-continuous models.

## 1 INTRODUCTION

Time series analysis and modeling are ubiquitous in a wide range of fields, including industry, medicine, and climate science. The variety, heterogeneity and increasing number of deployed sensors, raise new challenges when dealing with real-world problems for which current methods often fail. For example, data are frequently irregularly sampled, contain missing values, or are unaligned when collected from distributed sensors (Schulz and Stattegger, 1997; Clark and Bjørnstad, 2004). Recent advancements in deep learning have significantly improved state-of-the-art performance in both data imputation (Cao et al., 2018; Du et al., 2023) and forecasting tasks (Zeng et al., 2022; Nie et al., 2022). Many state-of-the-art models, such as transformers, have been primarily designed for dense and regular grids (Wu et al., 2021; Nie et al., 2022; Du et al., 2023). They struggle to handle irregular data and often suffer from significant performance degradation (Chen et al., 2001; Kim et al., 2019).

Our objective is to explore alternatives to SOTA transformers able to handle, in a unified framework, imputation and forecasting tasks for irregularly, arbitrarily sampled, and unaligned time series sources. Time-dependent continuous models (Rasmussen and Williams, 2006; Garnelo et al., 2018; Rubanova et al., 2019) offer such an alternative. However, until now, their performance has lagged significantly behind that of models designed for regular discrete grids. A few years ago, implicit neural representations (INRs) emerged as a powerful tool for representing images as continuous functions of spatial coordinates (Sitzmann et al., 2020; Tancik et al., 2020) with recent new applications such as image generation (Dupont et al., 2022) or even modeling dynamical systems (Yin et al., 2023).

In this work, we leverage the potential of conditional INR models within a meta-learning approach to introduce TimeFlow: a unified framework designed for modeling continuous time series and addressing imputation and forecasting tasks with irregular and unaligned observations. Our key contributions are the following:

- We propose a novel framework that excels in modeling time series as continuous functions of time, accepting arbitrary time step inputs, thus enabling the handling of irregular and unaligned time series for both imputation and forecasting tasks. This is one of the very first attempts to adapt INRs that enables efficient handling of both imputation and forecasting tasks within a unified framework. The methodology which leverages the synergy

between the model components, evidenced in the context of this application, is a pioneering contribution to the field.

- We conducted an extensive comparison with state-of-the-art continuous and discrete models. It demonstrates that our approach outperforms continuous and discrete SOTA deep learning approaches for imputation. As for long-term forecasting, it outperforms existing continuous models both on regular and irregular samples. It is on par with SOTA discrete models on regularly sampled time series while allowing for a much greater flexibility for irregular samplings, allowing to cope with situations where discrete models fail. Furthermore, we prove that our method effortlessly handles previously unseen time series and new time windows, making it well-suited for real-world applications.

## 2 RELATED WORK

**Discrete methods for time series imputation and forecasting.** Recently, Deep Learning (DL) methods have been widely used for both time series imputation and forecasting. For imputation, BRITS (Cao et al., 2018) uses a bidirectional recurrent neural network (RNN). Alternative frameworks were later explored, e.g., GAN-based (Luo et al., 2018; 2019; Liu et al., 2019), VAE-based (Fortuin et al., 2020), diffusion-based (Tashiro et al., 2021), matrix factorization-based (TIDER, Liu et al., 2023) and transformer-based (SAITS, Du et al., 2023) approaches. These methods cannot handle irregular time series. In situations involving multiple sensors, such as those placed at different locations, incorporating new sensors necessitates retraining the entire model, thereby limiting their usability. For forecasting, most recent DL SOTA models are based on transformers. Initial approaches apply plain transformers directly to the series, each token being a series element (Zhou et al., 2021; Liu et al., 2022; Wu et al., 2021; Zhou et al., 2022). These transformers may underperform linear models as shown in (Zeng et al., 2022). PatchTST (Nie et al., 2022) significantly improved transformers SOTA performance by considering sub-series as tokens of the series. However, all these models cannot handle properly irregularly sampled look-back windows.

**Continuous methods for time series.** Gaussian Processes (Rasmussen and Williams, 2006) have been a popular family of methods for modeling time series as continuous functions. They require choosing an appropriate kernel (Corani et al., 2021) and may suffer limitations in large dimensions settings. Neural Processes (NPs) (Garnelo et al., 2018; Kim et al., 2019) parameterize Gaussian processes through an encoder-decoder architecture leading to more computationally efficient implementations. NPs have been used to model simple signals for imputation and forecasting tasks, but struggle with more complex signals. Bilos et al. (2023) parameterizes a Gaussian Process through a diffusion model, but the model has difficulty adapting to a large number of timestamps. Other approaches such as Brouwer et al. (2019) and Rubanova et al. (2019) model time series continuously with latent ordinary differential equations. mTAN (Shukla and Marlin, 2021) a transformer model uses an attention mechanism to impute irregular time series. While these approaches have shown significant progress in continuous modeling for time series, their performances on regularly spaced grids are inferior compared to the aforementioned discrete models and they lack extrapolation capability when dealing with complex dynamics.

**Implicit neural representations.** The recent development of implicit neural representations (INRs) has led to impressive results in computer vision (Sitzmann et al., 2020; Tancik et al., 2020; Fathony et al., 2021; Mildenhall et al., 2021). INRs can represent data as a continuous function, which can be queried at any coordinate. While they have been applied in other fields such as physics (Yin et al., 2023) and meteorology (Huang and Hoefler, 2023), there has been limited research on INRs for time series analysis. Prior works (Fons et al., 2022; Jeong and Shin, 2022) focused on time series generation for data augmentation and on time series encoding for reconstruction but are limited by their fixed grid input requirement. DeepTime (Woo et al., 2022) is the closest work to our contribution. DeepTime learns a set of basis INR functions from a training set of multiple time series and combines them using a Ridge regressor. This regressor allows it to adapt to new time series. It has been designed for forecasting only. The original version cannot handle imputation properly and was adapted to do so for our comparisons. In our experiments, we will demonstrate that TimeFlow significantly outperforms DeepTime in imputation and also in forecasting tasks when dealing with missing values in the look-back window. TimeFlow also shows a slight advantage over DeepTime in forecasting regularly sampled series.

## 3 THE TIMEFLOW FRAMEWORK

### 3.1 PROBLEM SETTING

We aim to develop a unified framework for time series imputation and forecasting that reduces dependency on a fixed sampling scheme for time series. We introduce the following notations for both tasks. During training, in the imputation setting, we have access to time series in an observation set denoted as $\mathcal{T}_{in}$, which is a subset of the complete time series observation set $\mathcal{T}$. In the forecasting setting, we observe time series within a limited past time grid, referred to as the 'look-back window' and denoted as $\mathcal{T}_{in}$ (a subset of $\mathcal{T}$), as well as a future grid, the 'horizon', denoted as $\mathcal{T}_{out}$ (also a subset of $\mathcal{T}$). At test time, in both cases, and given an observed subset $\mathcal{T}_{in}^*$ included in a possibly new temporal window $\mathcal{T}^*$, our objective is to infer the time series values within $\mathcal{T}^*$.

### 3.2 KEY COMPONENTS

Our framework is articulated around three key components. (i) **INR-based time–continuous functions**: a time series $x$ is represented by a time-continuous function $f\colon t \in \mathbb{R}_+ \to f(t) \in \mathbb{R}^d$ that can be queried at any time $t$. For that, we employ implicit neural representations (INRs), which are neural networks capable of learning a parameterized continuous function $f_\theta$ from discrete data by minimizing the reconstruction loss between observed data and network's outputs. (ii) **Conditional INRs with modulations**: An INR can represent only one function, whether it's an image or a time series. To effectively represent a collection of time series $(x^{(j)})_j$ using INRs, we improve their encoding by incorporating per-sample modulations, which we denote as $\psi^{(j)}$. These modulations condition the

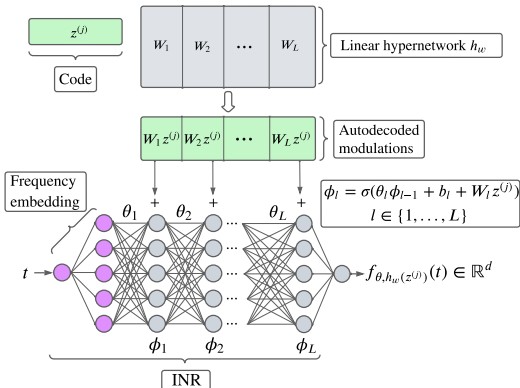

Figure 1: Overview of TimeFlow architecture. Forward pass to approximate the time series $x^{(j)}$. $\sigma$ stands for the ReLU activation function.

parameters $\theta$ of the INRs. We use the notation $f_{\theta,\psi^{(j)}}$ to refer to the conditioned INR with the modulations $\psi^{(j)}$. (iii) **Optimization-based encoding**: the conditioning modulation parameters $\psi^{(j)}$ are calculated as a function of codes $z^{(j)}$ that represent the individual sample series. We acquire these codes $z^{(j)}$ through a meta-learning optimization process using an auto-decoding strategy. Notably, auto-decoding has been found to be more efficient for this purpose than set encoders (Kim et al., 2019). In the following sections, we will elaborate on each component of our method. Given that the choices made for each component and the methodology developed to enhance their synergy are essential aspects, we provide a discussion of the various choices involved in Section 3.4.

**INR-based time-continuous functions.** We implement our INR with Fourier features and a feed-forward network (FFN) with ReLU activations, i.e. for a time coordinate $t \in \mathcal{T}$, the output of the INR $f_\theta$ is given by $f_\theta(t) = \text{FFN}(\gamma(t))$. The Fourier Features $\gamma(\cdot)$ are a frequency embedding of the time coordinates used to capture high-frequencies (Tancik et al., 2020; Mildenhall et al., 2021). In our case, we chose $\gamma(t) := (\sin(\pi t), \cos(\pi t), \cdots, \sin(2^{N-1}\pi t), \cos(2^{N-1}\pi t))$, with $N$ the number of fixed frequencies. For an INR with $L$ layers, the output is computed as follows: (i) we get the frequency embedding $\phi_0 = \gamma(t)$, (ii) we update the hidden states according to $\phi_l = \text{ReLU}(\theta_l \phi_{l-1} + b_l)$ for $l = 1, \ldots, L$, (iii) we project onto the output space $f_\theta(t) = \theta_{L+1}\phi_L + b_{L+1}$.

**Conditional INRs with modulations.** As indicated, sample conditioning of the INR is performed through modulations of its parameters. In order to adapt rapidly the model to new samples, the conditioning should rely only on a small number of the INR parameters. This is achieved by modifying only the biases of the INR through the introduction of an additional bias term $\psi_l^{(j)}$ for each layer $l$, also known as *shift modulation*. To further limit the versatility of the conditioning, we generate the instance modulations $\psi^{(j)}$ from compact codes $z^{(j)}$ through a linear hypernetwork $h$ with parameters $w$, i.e., $\psi^{(j)} = h_w(z^{(j)})$. Consequently, the approximation of a time series $x^{(j)}$, denoted

globally as $f_{\theta, h_w(z^{(j)})}$, will depend on shared parameters $\theta$ and $w$ that are common among all the INRs involved in modeling the series family and on the code $z^{(j)}$ specific to series $x^{(j)}$. The output of the $l$-th layer of the modulated INR is given by $\phi_l = \text{ReLU}(\theta_l \phi_{l-1} + b_l + \psi_l^{(j)})$, where $\psi_l^{(j)} = W_l z^{(j)}$, and $w := (W_l)_{l=1}^L$ are the parameters of the hypernetwork $h_w$. This design enables gathering information across samples into the common parameters of the INR and hypernetwork, while the codes contain only specific information about their respective time-series samples. The architecture is illustrated in Figure 1.

**Optimization-based encoding.** We condition the INR using the data from $\mathcal{T}_{in}$, and learn the shared INR and hypernetwork parameters $\theta$ and $w$ using $\mathcal{T}_{in}$ for both imputation and forecasting, and $\mathcal{T}_{out}$ for forecasting only. We achieve the conditioning on $\mathcal{T}_{in}$ by optimizing the codes $z^{(j)}$ through gradient descent. The joint optimization of the codes and common parameters is challenging. In TimeFlow, it is achieved through a meta-learning approach, adapted from Dupont et al. (2022) and Zintgraf et al. (2019). The objective is to learn shared parameters so that the code $z^{(j)}$ can be

---

**Algorithm 1:** TimeFlow Training

**while** *no convergence* **do**

  Sample batch $\mathcal{B}$ of data $(x^{(j)})_{j \in \mathcal{B}}$;

  Set codes to zero $z^{(j)} \leftarrow 0, \forall j \in \mathcal{B}$ ;

  // inner loop for encoding:

  **for** $j \in \mathcal{B}$ *and step* $\in \{1, ..., K\}$ **do**

    $z^{(j)} \leftarrow z^{(j)} - \alpha \nabla_{z^{(j)}} \mathcal{L}_{\mathcal{T}_{in}^{(j)}}(f_{\theta, h_w(z^{(j)})}, x^{(j)})$;

  // outer loop step:

  $[\theta, w] \leftarrow [\theta, w] -$

  $\eta \nabla_{[\theta,w]} \frac{1}{|\mathcal{B}|} \sum_{j \in \mathcal{B}} [\mathcal{L}_{\mathcal{T}_{in}^{(j)}}(f_{\theta, h_w(z^{(j)})}, x^{(j)}) +$

  $\lambda \mathcal{L}_{\mathcal{T}_{out}^{(j)}}(f_{\theta, h_w(z^{(j)})}, x^{(j)})]$ ;

---

adapted in just a few gradient steps for a new series $x^{(j)}$. For training, we perform parameter optimization at two levels: the inner-loop and the outer-loop. The inner-loop adapts the code $z^{(j)}$ to condition the network on the set $\mathcal{T}_{in}^{(j)}$, while the outer-loop updates the common parameters using $\mathcal{T}_{in}^{(j)}$ and also $\mathcal{T}_{out}^{(j)}$ for forecasting. We present our training optimization in Algorithm 1. At each training epoch and for each batch of data $\mathcal{B}$ composed of time series $x^{(j)}$ sampled from the training set, we first update individually the codes $z^{(j)}$ in the inner loop, before updating the common parameters in the outer loop using a loss over the whole batch. We introduce a parameter $\lambda$ to weight the importance of the loss over $\mathcal{T}_{out}$ w.r.t. the loss over $\mathcal{T}_{in}$ for the outer-loop. In practice, when $\mathcal{T}_{out}$ exists, i.e. for forecasting, we set $\lambda = 1$ and $\lambda = 0$ otherwise. We use an MSE loss over the observations grid $\mathcal{L}_{\mathcal{T}}(x_t, \tilde{x}_t) := \mathbb{E}_{t \sim \mathcal{T}}[(x_t - \tilde{x}_t)^2]$. We denote $\alpha$ and $\eta$ the learning rates of the inner- and outer-loop. Using $K = 3$ steps for training and testing is sufficient for our experiments thanks to the use of second-order meta-learning as explained in Section 3.4.

## 3.3 TIMEFLOW INFERENCE

During the inference process, we aim to infer the time series value for each timestamp in the dense grid $\mathcal{T}^{*(j)}$ based on the partial observation grid $\mathcal{T}_{in}^{*(j)} \subset \mathcal{T}^{*(j)}$. We can encounter two scenarios: (i) One where we observe the same time window as during training ($\mathcal{T}^{*(j)} = \mathcal{T}^{(j)}$) as in the

---

**Algorithm 2:** TimeFlow Inference with trained $\theta, w$

For the $j$-th series $(x^{(j)})$, set code to zero $z^{*(j)} \leftarrow 0$;

**for** *step* $\in \{1, ..., K\}$ **do**

  $z^{*(j)} \leftarrow z^{*(j)} - \alpha \nabla_{z^{*(j)}} \mathcal{L}_{\mathcal{T}_{in}^{*(j)}}(f_{\theta, h_w(z^{*(j)})}, x_t)$

Query $f_{\theta, h_w(z^{*(j)})}(t)$ for any $t \in \mathcal{T}^{*(j)}$

---

imputation setting in Section 4.1. (ii) One, where we are dealing with a newly observed time window ($\mathcal{T}^{*(j)} \neq \mathcal{T}^{(j)}$), as in the forecasting setting in Section 4.2. At inference, the parameters $\theta$ and $w$ are kept fixed to their final training values. We optimize the individual parameters $z^{*(j)}$ based on the newly observed grid $\mathcal{T}_{in}^{*(j)}$ using the $K$ inner-steps of the meta-learning algorithm as described in Algorithm 2. We are then in position to query $f_{\theta, h_w(z^{*(j)})}(t)$ for any given timestamp $t \in \mathcal{T}^{*(j)}$.

## 3.4 DISCUSSION ON IMPLEMENTATION CHOICES

As indicated before, adapting the components and enhancing their synergy for the tasks of imputation and forecasting is not trivial and requires careful choices. We conducted several ablation studies to provide a comprehensive examination of key implementation choices of our framework.

Our findings can be summarized as follows. **Choice of INR**: An FFN with Fourier Features outperformed other popular INRs for the tasks considered in this study. Unlike SIREN, which does not explicitly incorporate frequencies but uses sine activation functions, the Fourier features network can more effectively capture a broader range of frequencies, especially at low sampling rates. This is crucial for accurately capturing high frequencies in sparsely observed time series. Our experiments, detailed in Section A.2.1 and Table 4, demonstrate this superiority across various datasets. **Choice of encoding / meta-learning**: TimeFlow with a set encoder for learning the compact conditioning codes $z$ in place of the auto-decoding strategy used here, proved much less effective on complex datasets. This is further elaborated in Section A.2.4 and Table 11. Additionally, replacing the 2nd-order optimization for a 1st-order one, such as REPTILE, led to unstable training, as shown in Table 10. **Choice of modulations**: Complexifying the modulation by introducing scaling parameters in addition to shift parameters did not provide performance gains. Our experiments on the *Electricity* dataset, detailed in Section A.2.5 and Table 12, indicate that shift-only modulation is more efficient.

For TimeFlow, across all experiments, we used a code dimension of 128, an FFN with a depth of 5 and a width of 256, and 64 Fourier features. We used 3 inner steps and a learning rate of 0.01 for the inner-loop, and a learning rate of $5 \times 10^{-4}$ for the outer-loop. We performed a comprehensive analysis to understand notably the **influence of the $z$ dimension**: a latent code dimension of 128 was suitable for our tasks; this is supported by results in Section A.2.2 and Table 5 - and the **influence of the number of inner steps**: using 3 inner steps for training and inference struck a favorable balance between reconstruction capabilities and computational efficiency, as detailed in Section A.2.3.

## 4 EXPERIMENTS

We conducted a comprehensive evaluation of our TimeFlow framework across three different tasks, comparing its performance to state-of-the-art continuous and discrete baseline methods. In Section 4.1, we assess TimeFlow's capabilities to impute sparsely observed time series under various sampling rates. Section 4.2 focuses on long-term forecasting, where we evaluate TimeFlow over standard long-term forecasting horizons. In Section 4.3, we tackle a challenging task forecasting with incomplete look-back windows, thus combining the challenges of imputation and forecasting. This demonstrates TimeFlow's versatility and performance.

**Datasets.** We tested our framework on three extensive multivariate datasets where a single phenomenon is measured at multiple locations over time, namely *Electricity*, *Traffic* and *Solar*. They are commonly used in the time series literature and are described in Appendix B.

### 4.1 IMPUTATION

We consider the classical imputation setting where $n$ time series are partially observed over a given time window. Using our approach, we can predict for each time series the value at any timestamp $t$ in that time window based on partial observations.

**Setting.** For a time series $x^{(j)}$, we denote the set of observed points as $\mathcal{T}_{in}^{(j)}$ and the ground truth set of points as $\mathcal{T}^{(j)}$. The observed time grids may be irregularly spaced and may differ across the different time series $(\mathcal{T}_{in}^{(j_1)} \neq \mathcal{T}_{in}^{(j_2)}, \forall j_1 \neq j_2)$. The model is trained for each $x^{(j)}$ following Algorithm 1. Then, we aim to infer for any unobserved $t \in \mathcal{T}^{(j)}$ the missing value $x_t^{(j)}$ conditioned on $\mathcal{T}_{in}^{(j)}$ according to Algorithm 2. For this imputation task, the TimeFlow training and inference procedures are detailed in Section 3 and illustrated in Figure 2. For comparison with the SOTA imputation baselines, we assume that the ground truth time grid is the same for each sample. The subsampling rate $\tau$ is define as the rate of observed values.

**Baselines.** We compare TimeFlow with various baselines, including discrete imputation methods, such as CSDI (Tashiro et al., 2021), SAITS (Du et al., 2023), BRITS (Cao et al., 2018), and TIDER (Liu et al., 2023), and continuous ones, such as Neural Process (NP, Garnelo et al., 2018), mTAN (Shukla and Marlin, 2021), and DeepTime with slight adjustments (Woo et al., 2022) (details cf. Appendix C.3). For each dataset, we divide the series into five independent time windows (consisting of 2000 timestamps for *Electricity* and *Traffic*, and 10,000 timestamps for *Solar*), perform imputation

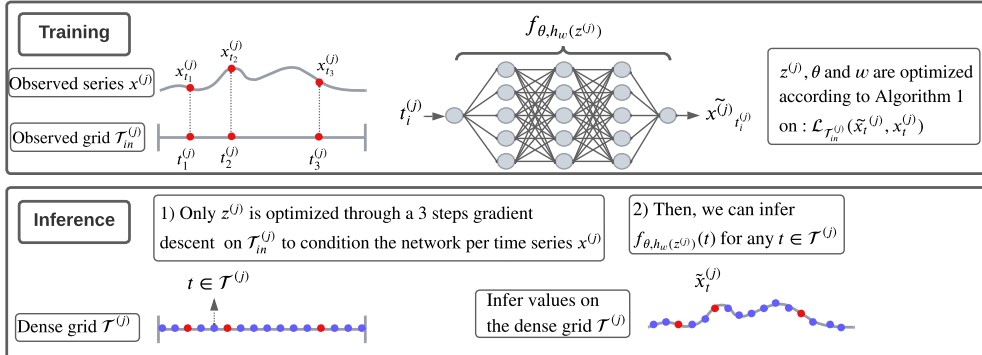

Figure 2: Training and inference procedures of TimeFlow for imputation. (i) During training, for each time series $x^{(j)}$, our observations (red dots •) are restricted to the sparsely sampled grid, denoted as $\mathcal{T}_{in}^{(j)}$. (ii) During inference, our objective is to infer the values over the dense grids $\mathcal{T}^{(j)}$, on the unobserved data points (such as the blue dots • on the figure).

Table 1: Mean MAE imputation results on the missing grid only. Each time series is divided into 5 time windows onto which imputation is performed, and the performances are averaged over the 5 windows. In the table, $\tau$ stands for the subsampling rate, i.e. the proportion of observed points considered for each time window. Bold results are best, underlined results are second best. TimeFlow improvement represents the overall percentage improvement achieved by TimeFlow compared to the specific method being considered.

| | | Continuous methods | | | | Discrete methods | | | |
|---|---|---|---|---|---|---|---|---|---|
| | $\tau$ | TimeFlow | DeepTime | mTAN | Neural Process | CSDI | SAITS | BRITS | TIDER |
| Electricity | 0.05 | **0.324 ± 0.013** | 0.379 ± 0.037 | 0.575 ± 0.039 | 0.357 ± 0.015 | 0.462 ± 0.021 | 0.384 ± 0.019 | 0.329 ± 0.015 | 0.427 ± 0.010 |
| | 0.10 | **0.250 ± 0.010** | 0.333 ± 0.034 | 0.412 ± 0.047 | 0.417 ± 0.057 | 0.398 ± 0.072 | 0.308 ± 0.011 | 0.287 ± 0.015 | 0.399 ± 0.009 |
| | 0.20 | **0.225 ± 0.008** | 0.244 ± 0.013 | 0.342 ± 0.014 | 0.320 ± 0.017 | 0.341 ± 0.068 | 0.261 ± 0.008 | 0.245 ± 0.011 | 0.391 ± 0.010 |
| | 0.30 | **0.212 ± 0.007** | 0.240 ± 0.014 | 0.335 ± 0.015 | 0.300 ± 0.022 | 0.277 ± 0.059 | 0.236 ± 0.008 | 0.221 ± 0.008 | 0.384 ± 0.009 |
| | 0.50 | 0.194 ± 0.007 | 0.227 ± 0.012 | 0.340 ± 0.022 | 0.297 ± 0.016 | **0.168 ± 0.003** | 0.209 ± 0.008 | 0.193 ± 0.008 | 0.386 ± 0.009 |
| Solar | 0.05 | **0.095 ± 0.015** | 0.190 ± 0.020 | 0.241 ± 0.102 | 0.115 ± 0.015 | 0.374 ± 0.033 | 0.142 ± 0.016 | 0.165 ± 0.014 | 0.291 ± 0.009 |
| | 0.10 | **0.083 ± 0.015** | 0.159 ± 0.043 | 0.251 ± 0.081 | 0.114 ± 0.014 | 0.375 ± 0.038 | 0.124 ± 0.018 | 0.132 ± 0.015 | 0.276 ± 0.010 |
| | 0.20 | **0.072 ± 0.015** | 0.149 ± 0.020 | 0.314 ± 0.035 | 0.109 ± 0.016 | 0.217 ± 0.023 | 0.108 ± 0.014 | 0.109 ± 0.012 | 0.270 ± 0.010 |
| | 0.30 | **0.061 ± 0.012** | 0.135 ± 0.014 | 0.338 ± 0.05 | 0.108 ± 0.016 | 0.156 ± 0.002 | 0.100 ± 0.015 | 0.098 ± 0.012 | 0.266 ± 0.010 |
| | 0.50 | **0.054 ± 0.013** | 0.098 ± 0.013 | 0.315 ± 0.080 | 0.107 ± 0.015 | 0.079 ± 0.011 | 0.094 ± 0.013 | 0.088 ± 0.013 | 0.262 ± 0.009 |
| Traffic | 0.05 | 0.283 ± 0.016 | **0.246 ± 0.010** | 0.406 ± 0.074 | 0.318 ± 0.014 | 0.337 ± 0.045 | 0.293 ± 0.007 | 0.261 ± 0.010 | 0.363 ± 0.007 |
| | 0.10 | **0.211 ± 0.012** | 0.214 ± 0.007 | 0.319 ± 0.025 | 0.288 ± 0.018 | 0.288 ± 0.017 | 0.237 ± 0.006 | 0.245 ± 0.009 | 0.362 ± 0.006 |
| | 0.20 | **0.168 ± 0.006** | 0.216 ± 0.006 | 0.270 ± 0.042 | 0.271 ± 0.011 | 0.269 ± 0.017 | 0.197 ± 0.005 | 0.224 ± 0.008 | 0.361 ± 0.006 |
| | 0.30 | **0.151 ± 0.007** | 0.172 ± 0.008 | 0.251 ± 0.006 | 0.259 ± 0.012 | 0.240 ± 0.037 | 0.180 ± 0.006 | 0.197 ± 0.007 | 0.355 ± 0.006 |
| | 0.50 | **0.139 ± 0.007** | 0.171 ± 0.005 | 0.278 ± 0.040 | 0.240 ± 0.021 | 0.144 ± 0.022 | 0.160 ± 0.008 | 0.161 ± 0.060 | 0.354 ± 0.007 |
| TimeFlow improvement | / | | 24.14 % | 50.53 % | 31.61 % | 36.12 % | 20.33 % | 18.90 % | 53.40 % |

on each time window and average the performance to obtain robust results. We evaluate the quality of the models for different subsampling rates, from the easiest $\tau = 0.5$ to the most difficult $\tau = 0.05$.

**Results.** We show in Table 1 that TimeFlow outperforms both discrete and continuous models across almost all $\tau$s for the given datasets. The relative improvements of TimeFlow over baselines are significant, ranging from 15% to 50%. Especially for the lowest sampling rate $\tau = 0.05$, TimeFlow outperforms all discrete baselines, demonstrating the advantages of continuous modeling. Additionally, it achieves lower imputation errors compared to continuous models in all but one cases. Qualitatively, we see on example series in Figure 3 that our model shows significant imputation capabilities, with on a subsampling rate at $\tau = 0.1$ on the *Electricity* dataset. It captures well different frequencies and amplitudes in a challenging case (sample 35), although it underestimates the amplitude of some peaks. In a more challenging scenario (sample 25), where the series exhibit additional trend changes and frequency variations within the data, TimeFlow correctly imputes most timestamps, outperforming BRITS, which is the best-performing method for the *Electricity* dataset.

**Imputation on previously unseen time series.** In more practical scenarios, such as cases involving the installation of new sensors, we often encounter new time series originating from the same

underlying phenomenon. In such instances, it becomes crucial to make inferences for these previously unseen time series. Thanks to efficient adaptation in latent space, our model can easily be applied to these new time series (as shown in Appendix C.2, Table 15), contrasting with SOTA methods like SAITS and BRITS, which require full model retraining on the whole set of time series.

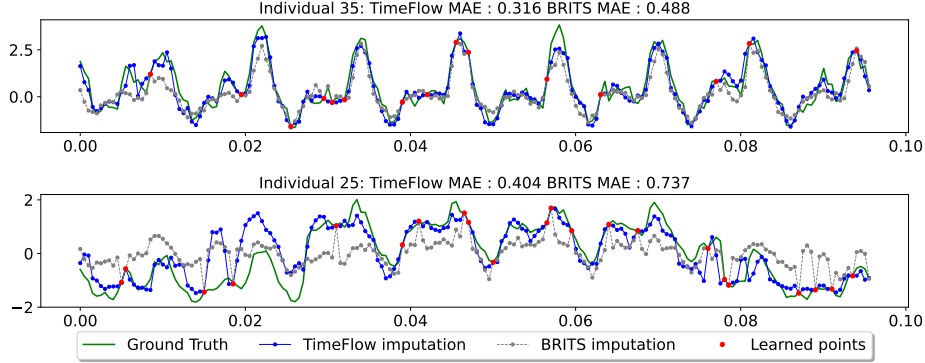

Figure 3: *Electricity dataset*. TimeFlow imputation (blue line) and BRITS imputation (gray line) with 10% of known point (red points) on the eight first days of samples 35 (top) and 25 (bottom).

## 4.2 FORECASTING

In this section, we are interested in the conventional long-term forecasting scenario. It consists in predicting the phenomenon in a specific future period, the horizon, based on the history of a limited past period, the look-back window. The forecaster is trained on a set of $n$ observed time series for a given time window (train period) and tested on new time windows.

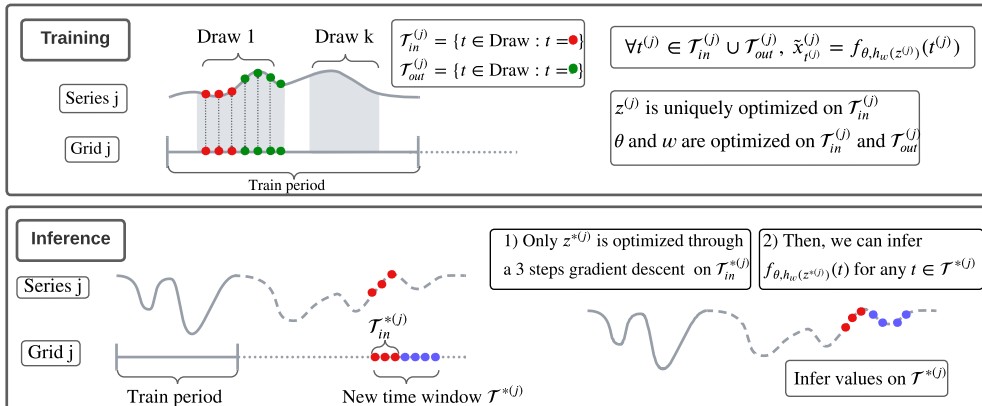

Figure 4: Training and inference procedure of TimeFlow for forecasting. (i) During training (top–figure), for each time series $x^{(j)}$, we observe some look-back window/horizon drawing pairs in the trained period. TimeFlow is trained with Algorithm 1 to predict all observed timestamps in this drawing pairs while being conditioned by the observed look-back window. (ii) Once TimeFlow is optimized, the objective during inference (bottom-figure) is to infer the horizon over new time windows (blue dots •) while being conditioned by the newly observed look-back window (red dots •).

**Setting** For a given time series $x^{(j)}$, $\mathcal{T}_{in}^{(j)}$ denotes the look-back window and $\mathcal{T}_{out}^{(j)}$ the horizon of $H$ points. During training, at each epoch, we train $f_{\theta,h_w(z^{(j)})}$ following Algorithm 1 with randomly drawn pairs of look-back window and horizon $(\mathcal{T}_{in}^{(j)} \cup \mathcal{T}_{out}^{(j)})_{j \in \mathcal{B}}$ within the observed train period. Then, for a new time window $\mathcal{T}^{*(j)}$, given a look-back window $\mathcal{T}_{in}^{*(j)}$ we forecast future values any $t \in \mathcal{T}^{*(j)}$, the horizon interval, following Algorithm 2. We illustrate the training and inference of TimeFlow for the forecasting task in Figure 4.

**Baselines.** To evaluate the quality of our model in long-term forecasting, we compare it to the discrete baselines PatchTST (Nie et al., 2022), DLinear (Zeng et al., 2022), AutoFormer (Wu et al., 2021), and Informer (Zhou et al., 2021). We also include continuous baselines DeepTime and Neural Process (NP). In Table 2, we present the forecasting results for standard horizons in long-term forecasting: $H \in \{96, 192, 336, 720\}$. The look-back window length is fixed to 512.

Table 2: Mean MAE forecast results averaged over different time windows. Each time, the model is trained on one time window and tested on the others (there are 2 windows for *SolarH* and 5 for *Electricity* and *Traffic*). $H$ stands for the horizon. Bold results are best, and underlined results are second best. TimeFlow improvement represents the overall percentage improvement achieved by TimeFlow compared to the specific method being considered.

| | | Continuous methods | | | Discrete methods | | | |
|---|---|---|---|---|---|---|---|---|
| | $H$ | TimeFlow | DeepTime | Neural Process | Patch-TST | DLinear | AutoFormer | Informer |
| Electricity | 96 | $\underline{0.228 \pm 0.028}$ | $0.244 \pm 0.026$ | $0.392 \pm 0.045$ | $\mathbf{0.221 \pm 0.023}$ | $0.241 \pm 0.030$ | $0.546 \pm 0.277$ | $0.603 \pm 0.255$ |
| | 192 | $\underline{0.238 \pm 0.020}$ | $0.252 \pm 0.019$ | $0.401 \pm 0.046$ | $\mathbf{0.229 \pm 0.020}$ | $0.252 \pm 0.025$ | $0.500 \pm 0.190$ | $0.690 \pm 0.291$ |
| | 336 | $\underline{0.270 \pm 0.031}$ | $0.284 \pm 0.034$ | $0.434 \pm 0.076$ | $\mathbf{0.251 \pm 0.027}$ | $0.288 \pm 0.038$ | $0.523 \pm 0.188$ | $0.736 \pm 0.271$ |
| | 720 | $\underline{0.316 \pm 0.055}$ | $0.359 \pm 0.051$ | $0.607 \pm 0.150$ | $\mathbf{0.297 \pm 0.039}$ | $0.365 \pm 0.059$ | $0.631 \pm 0.237$ | $0.746 \pm 0.265$ |
| SolarH | 96 | $\mathbf{0.190 \pm 0.013}$ | $\underline{0.190 \pm 0.020}$ | $0.221 \pm 0.048$ | $0.262 \pm 0.070$ | $0.208 \pm 0.014$ | $0.245 \pm 0.045$ | $0.248 \pm 0.022$ |
| | 192 | $\mathbf{0.202 \pm 0.020}$ | $\underline{0.204 \pm 0.028}$ | $0.244 \pm 0.048$ | $0.253 \pm 0.051$ | $0.217 \pm 0.022$ | $0.333 \pm 0.107$ | $0.270 \pm 0.031$ |
| | 336 | $\underline{0.209 \pm 0.017}$ | $\mathbf{0.199 \pm 0.026}$ | $0.240 \pm 0.006$ | $0.259 \pm 0.071$ | $0.217 \pm 0.026$ | $0.334 \pm 0.079$ | $0.328 \pm 0.048$ |
| | 720 | $\mathbf{0.218 \pm 0.041}$ | $\underline{0.229 \pm 0.024}$ | $0.403 \pm 0.147$ | $0.267 \pm 0.064$ | $0.249 \pm 0.034$ | $0.351 \pm 0.055$ | $0.337 \pm 0.037$ |
| Traffic | 96 | $\underline{0.217 \pm 0.032}$ | $0.228 \pm 0.032$ | $0.283 \pm 0.027$ | $\mathbf{0.203 \pm 0.037}$ | $0.228 \pm 0.033$ | $0.319 \pm 0.059$ | $0.372 \pm 0.078$ |
| | 192 | $\underline{0.212 \pm 0.028}$ | $0.220 \pm 0.022$ | $0.292 \pm 0.024$ | $\mathbf{0.197 \pm 0.030}$ | $0.221 \pm 0.023$ | $0.368 \pm 0.057$ | $0.511 \pm 0.247$ |
| | 336 | $\underline{0.238 \pm 0.034}$ | $0.245 \pm 0.038$ | $0.305 \pm 0.039$ | $\mathbf{0.222 \pm 0.039}$ | $0.250 \pm 0.040$ | $0.434 \pm 0.061$ | $0.561 \pm 0.263$ |
| | 720 | $\underline{0.279 \pm 0.050}$ | $0.290 \pm 0.052$ | $0.339 \pm 0.038$ | $\mathbf{0.269 \pm 0.057}$ | $0.300 \pm 0.057$ | $0.462 \pm 0.062$ | $0.638 \pm 0.067$ |
| TimeFlow improvement | | / | 3.74 % | 29.06 % | 3.23 % | 6.92 % | 42.09 % | 48.57 % |

**Results.** The results in Table 2 show that our approach ranks in the top two across all datasets and horizons and is the overall best continuous method. TimeFlow's performance is comparable to the current SOTA model PatchTST, with only 2% relative difference. Moreover, TimeFlow shows consistent results across the three datasets, whereas the other best discrete and continuous baselines, i.e. PatchTST and DeepTime, performance drops for some datasets. We also note that, despite the great performance of the SOTA PatchTST, other transformer-based baselines (discrete methods in Table 2) perform poorly. We provide a detailed insight on these results in Appendix D.1. Overall, although this evaluation setting favors discrete methods because the time series are observed at evenly distributed time steps, TimeFlow consistently performs as well as PatchTST and outperforms all the other methods, whether discrete or continuous. It is the first time that a continuous model has achieved the same level of performance as discrete methods within their specific setting.

**Forecasting on previously unseen time series.** TimeFlow considers that the series observed at different locations are independent, similar to PatchTST, NP, and DeepTime. This allows it to generalize to previously unseen time series from the same phenomenon. Note that this is not the case for most discrete methods. We show in Appendix D.4, Table 21 that TimeFlow is able to generalize to previously unseen time series with no significant performance drop.

### 4.3 CHALLENGING TASK: FORECAST WHILE IMPUTING INCOMPLETE LOOK-BACK WINDOWS

In real-world scenarios, it is common to encounter missing or irregularly sampled series when making predictions on new time windows (Cinar et al., 2018; Tang et al., 2020). Continuous methods can handle these cases, as they are designed to accommodate irregular sampling within the look-back window. In this section, we formulate a task to simulate these real-world scenarios. It's worth noting that this task is often encountered in practice but is rarely considered in the DL literature.

**Setting and baselines.** This scenario is similar to the forecast setting in Section 4.2 and illustrated in Figure 4. The difference is that during inference, the look-back window is subsampled at a rate $\tau$ smaller than the one used for the training phase. This simulates a situation with missing observations in the look back window. Consequently, two distinct tasks emerge during the inference phase: imputing missing points within the sparsely observed look-back window, and forecasting over the horizon with this degraded context. In Table 3, we compare to the two other continuous baselines, DeepTime and NP on *Electricity* and *Traffic* for different $\tau$s and horizons.

Table 3: MAE results for forecasting with missing values in the look-back window. $\tau$ stands for the percentage of observed values in the look-back window. Best results are in bold. TimeFlow improvement represents the overall percentage improvement (for each task) achieved by TimeFlow compared to the specific method being considered.

| | $H$ | $\tau$ | TimeFlow | | DeepTime | | Neural Process | |
|---|---|---|---|---|---|---|---|---|
| | | | Imputation error | Forecast error | Imputation error | Forecast error | Imputation error | Forecast error |
| Electricity | 96 | 0.5 | **0.151 ± 0.003** | **0.239 ± 0.013** | 0.209 ± 0.004 | 0.270 ± 0.019 | 0.460 ± 0.048 | 0.486 ± 0.078 |
| | | 0.2 | **0.208 ± 0.006** | **0.260 ± 0.015** | 0.249 ± 0.006 | 0.296 ± 0.023 | 0.644 ± 0.079 | 0.650 ± 0.095 |
| | | 0.1 | **0.272 ± 0.006** | **0.295 ± 0.016** | 0.284 ± 0.007 | 0.324 ± 0.026 | 0.740 ± 0.083 | 0.737 ± 0.106 |
| | 192 | 0.5 | **0.149 ± 0.004** | **0.235 ± 0.011** | 0.204 ± 0.004 | 0.265 ± 0.018 | 0.461 ± 0.045 | 0.498 ± 0.070 |
| | | 0.2 | **0.209 ± 0.006** | **0.257 ± 0.013** | 0.244 ± 0.007 | 0.290 ± 0.023 | 0.601 ± 0.075 | 0.626 ± 0.101 |
| | | 0.1 | **0.274 ± 0.010** | **0.289 ± 0.016** | 0.282 ± 0.007 | 0.315 ± 0.025 | 0.461 ± 0.045 | 0.724 ± 0.090 |
| Traffic | 96 | 0.5 | **0.180 ± 0.016** | **0.219 ± 0.026** | 0.272 ± 0.028 | 0.243 ± 0.030 | 0.436 ± 0.025 | 0.444 ± 0.047 |
| | | 0.2 | **0.239 ± 0.019** | **0.243 ± 0.027** | 0.335 ± 0.026 | 0.293 ± 0.027 | 0.596 ± 0.049 | 0.597 ± 0.075 |
| | | 0.1 | **0.312 ± 0.020** | **0.290 ± 0.027** | 0.385 ± 0.025 | 0.344 ± 0.027 | 0.734 ± 0.102 | 0.731 ± 0.132 |
| | 192 | 0.5 | **0.176 ± 0.014** | **0.217 ± 0.017** | 0.241 ± 0.027 | 0.234 ± 0.021 | 0.477 ± 0.042 | 0.476 ± 0.043 |
| | | 0.2 | **0.233 ± 0.017** | **0.236 ± 0.021** | 0.286 ± 0.027 | 0.276 ± 0.020 | 0.685 ± 0.109 | 0.678 ± 0.108 |
| | | 0.1 | **0.304 ± 0.019** | **0.277 ± 0.021** | 0.331 ± 0.025 | 0.324 ± 0.021 | 0.888 ± 0.178 | 0.877 ± 0.174 |
| TimeFlow improvement | | | / | / | 18.97 % | 11.87 % | 61.88 % | 58.41 % |

**Results.** In Table 3, the results show that TimeFlow consistently outperforms other methods in imputation and forecasting for every scenarios. When comparing with the complete look-back windows observations scenario from Table 2, one observes that at a 0.5 sampling rate, TimeFlow presents only a slight reduction in performance, whereas other baseline methods experience more significant drops. For instance, when we compare forecast results between a complete window and a $\tau = 0.5$ subsampled window for *Electricity* with a forecasting horizon of $H = 96$, TimeFlow's error increases by a mere 4.6% (from 0.228 to 0.239). In contrast, DeepTime's error grows by over 10% (from 0.244 to 0.270), and NP experiences a rise of around 25% (from 0.392 to 0.486). For lower sampling rates, TimeFlow still delivers correct predictions. Qualitatively, we see on the series example in Figure 5 that despite observing only 10% of the look-back window, the model can correctly infer both the complete look-back window and the horizon. Both quantitative and qualitative results show the robustness and efficiency of TimeFlow on this particularly challenging setting.

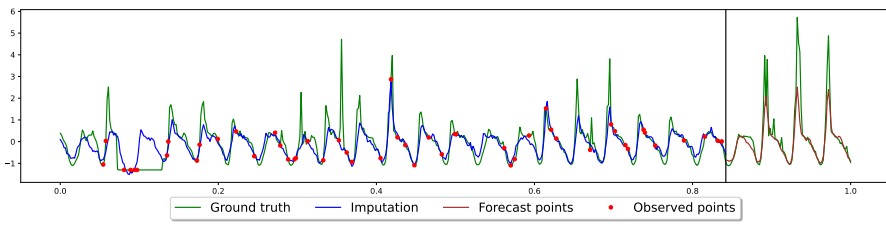

Figure 5: *Traffic dataset, sample 95*. In this figure, TimeFlow simultaneously imputes and forecasts at horizon 96 with a 10% partially observed look-back window of length 512.

## 5 CONCLUSION

We have introduced a unified framework for continuous time series modeling leveraging conditional INR and meta-learning. Our experiments have demonstrated superior performance compared to other continuous methods, and better or comparable results to SOTA discrete methods. One of the standout features of our framework is its inherent continuity and the ability to modulate the INR parameters. This unique flexibility lets TimeFlow effectively tackle a wide array of challenges, including forecasting in the presence of missing values, accommodating irregular time steps, and extending the trained model's applicability to previously unseen time series and new time windows. Our empirical results have shown TimeFlow's effectiveness in handling homogeneous multivariate time series. As a logical next step, extending TimeFlow's capabilities to address heterogeneous multivariate phenomena represents a promising direction for future research.

## REPRODUCTIBLITY STATEMENT

Our work is entirely reproducible, and all the references to the information in order to reproduce it are in this section.

**Code.**   The code for all our experiments is available at this link.

**Data.**   A subset of the processed data is available with the code in this link. The dataset description, processing and normalization are presented in Appendix B.

**Model.**   The model and the training details are presented in Section 3 and the hyperparameter selection is available in Appendix A.1.

**GPU.**   We used NVIDIA TITAN RTX 24Go single GPU to conduct all the experiments for our method, which is coded in PyTorch (Python 3.9.2).

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

# A  ARCHITECTURE DETAILS AND ABLATION STUDIES

## A.1  ARCHITECTURE DETAILS

For all imputation and forecasting experiments we choose the following hyperparameters :

- $z$ dimension: 128
- Number of layers: 5
- Hidden layers dimension: 256
- $\gamma(t) \in \mathbb{R}^{2 \times 64}$
- $z$ code learning rate ($\alpha$ in Algorithm 1): $10^{-2}$
- Hypernetwork and INR learning rate: $5 \times 10^{-4}$
- Number of steps in inner loop: $K = 3$
- Number of epochs: $4 \times 10^4$
- Batch size: 64

It is worth noting that the hyperparameters mentioned above remain consistent across all experiments conducted in the paper. We chose to maintain a fixed set of hyperparameters for our model, while other imputation and forecasting approaches commonly fine-tune hyperparameters based on a validation dataset. The obtained results exhibit high robustness across various settings, suggesting that the selected hyperparameters are already effective in achieving reliable outcomes.

## A.2  ABLATION STUDIES

### A.2.1  FOURIER FEATURES VS SIREN ON IMPUTATION TASK

**Baseline**  The SIREN network differs from the Fourier features network because it does not explicitly incorporate frequencies as input. Instead, it is a multi-layer perceptron network that utilizes sine activation functions. An adjustable parameter, denoted $\omega_0$, is multiplied with the input matrices of the preceding layers to capture a broader range of frequencies. For this comparison, we adopt the same hyperparameters described in Appendix A.1, selecting $\omega_0 = 30$ to align with Sitzmann et al. (2020). Furthermore, we set the learning rate of both the hypernetwork and the INR to $5 \times 10^{-5}$ to enhance training stability. In Table 4, we compare the imputation results obtained by the Fourier features network and the SIREN network, specifically focusing on the first time window from the *Electricity*, *Traffic* and *Solar* datasets.

Table 4: MAE imputation errors on the first time window of each dataset. Best results are bold.

|  | $\tau$ | TimeFlow | TimeFlow w SIREN |
|---|---|---|---|
| Electricity | 0.05 | **0.323** | 0.466 |
|  | 0.10 | **0.252** | 0.350 |
|  | 0.20 | **0.224** | 0.242 |
|  | 0.30 | **0.211** | 0.222 |
|  | 0.50 | **0.194** | 0.209 |
| Solar | 0.05 | **0.105** | 0.114 |
|  | 0.10 | **0.083** | 0.094 |
|  | 0.20 | **0.065** | 0.079 |
|  | 0.30 | **0.061** | 0.072 |
|  | 0.50 | **0.056** | 0.066 |
| Traffic | 0.05 | **0.292** | 0.333 |
|  | 0.10 | **0.220** | 0.252 |
|  | 0.20 | **0.168** | 0.191 |
|  | 0.30 | **0.152** | 0.163 |
|  | 0.50 | **0.141** | 0.154 |

**Results**  According to the results presented in Table 4, the Fourier features network outperforms the SIREN network in the imputation task on these datasets. Notably, the performance gap between the two network architectures are more pronounced at low sampling rates. This disparity can be attributed to the SIREN network's difficulty in accurately capturing high frequencies when the time series is sparsely observed. We hypothesize that the MLP with ReLU activations correctly learns the different frequencies of time series with multi-temporal patterns by switching on or off the Fourier embedding frequencies.

A.2.2 INFLUENCE OF THE LATENT CODE DIMENSION

The dimension of the latent code $z$ is a crucial parameter in our architecture. If it is too small, it underfits the timeseries. Consequently, this adversely affects the performance of both the imputation and forecasting tasks. On the other hand, if the dimension of $z$ is too large, it can lead to overfitting, hindering the model's ability to generalize to new data points.

**Baselines** To investigate the impact of $z$ dimensionality on the performance of TimeFlow, we conducted experiments on the three considered datasets, specifically focusing on the forecasting task. We varied the sizes of $z$ within $\{32, 64, 128, 256\}$. The other hyperparameters are set as presented in Appendix A.1. The obtained results for each $z$ dimension are summarized in Table 5.

Table 5: MAE error for different $z$ dimension.

|  | H | 32 | 64 | 128 | 256 |
|---|---|---|---|---|---|
| Electricity | 96 | $0.232 \pm 0.016$ | $0.222 \pm 0.017$ | $0.222 \pm 0.018$ | $\mathbf{0.215 \pm 0.019}$ |
|  | 192 | $0.245 \pm 0.020$ | $0.239 \pm 0.018$ | $\mathbf{0.230 \pm 0.026}$ | $0.233 \pm 0.017$ |
|  | 336 | $0.254 \pm 0.029$ | $0.244 \pm 0.028$ | $0.262 \pm 0.031$ | $\mathbf{0.243 \pm 0.032}$ |
|  | 720 | $0.295 \pm 0.027$ | $0.284 \pm 0.028$ | $0.303 \pm 0.041$ | $\mathbf{0.283 \pm 0.029}$ |
| SolarH | 96 | $0.182 \pm 0.009$ | $0.181 \pm 0.012$ | $\mathbf{0.179 \pm 0.003}$ | $0.225 \pm 0.047$ |
|  | 192 | $0.195 \pm 0.014$ | $0.195 \pm 0.016$ | $\mathbf{0.193 \pm 0.015}$ | $0.197 \pm 0.029$ |
|  | 336 | $\mathbf{0.181 \pm 0.011}$ | $0.182 \pm 0.011$ | $0.189 \pm 0.013$ | $0.183 \pm 0.012$ |
|  | 720 | $0.201 \pm 0.027$ | $\mathbf{0.199 \pm 0.025}$ | $0.209 \pm 0.029$ | $0.200 \pm 0.030$ |
| Traffic | 96 | $0.223 \pm 0.024$ | $0.215 \pm 0.028$ | $0.215 \pm 0.037$ | $\mathbf{0.210 \pm 0.033}$ |
|  | 192 | $0.214 \pm 0.018$ | $0.217 \pm 0.025$ | $0.206 \pm 0.023$ | $\mathbf{0.203 \pm 0.024}$ |
|  | 336 | $0.238 \pm 0.029$ | $0.231 \pm 0.029$ | $\mathbf{0.226 \pm 0.030}$ | $0.229 \pm 0.029$ |
|  | 720 | $0.272 \pm 0.040$ | $0.269 \pm 0.035$ | $\mathbf{0.259 \pm 0.038}$ | $0.262 \pm 0.040$ |

**Results** The results presented in Table 5 suggest that a $z$ dimension of 128 is a reasonable compromise but only optimal for some settings. Moreover, even though the choice of $z$ dimension seems important, it doesn't critically impact the MAE error for the forecasting task.

A.2.3 INFLUENCE OF THE NUMBER OF GRADIENT STEPS

As can be seen in Table 6, using three gradient steps at inference yield an inference time of less than 0.2 seconds. The latter can still be reduced by doing only one step at the cost of an increase in the forecasting error. As observed in Table 6, increasing the number of gradient steps above 3 steps during inference does not improve forecasting performance.

Table 6: Inference time (in seconds) and MAE error on the forecasting task on the *Electricity* dataset for a horizon of length 720, a look-back window of length 512, and a varying number of adaptation gradient steps. The statistics are computed over 10 runs using an NVIDIA TITAN RTX GPU.

| Gradient descent steps | 1 | 3 | 10 | 50 | 500 | 5000 |
|---|---|---|---|---|---|---|
| Inference time (s) | $0.109 \pm 0.003$ | $0.176 \pm 0.009$ | $0.427 \pm 0.031$ | $3.547 \pm 0.135$ | $17.722 \pm 0.536$ | $189.487 \pm 8.060$ |
| MAE | $0.351 \pm 0.038$ | $0.303 \pm 0.041$ | $0.300 \pm 0.040$ | $0.299 \pm 0.039$ | $0.302 \pm 0.038$ | $0.308 \pm 0.037$ |

Table 7: MAE error on the forecasting task using 1 inner-step during training and a varying number of adaptation gradient steps at inference. Best results are in bold and / symbol means that the MAE score is very high ($\geq 1$).

|  | H | 1 | 3 | 10 | 50 |
|---|---|---|---|---|---|
| Electricity | 96 | $\mathbf{0.244 \pm 0.017}$ | $0.246 \pm 0.017$ | $0.261 \pm 0.016$ | / |
|  | 192 | $\mathbf{0.253 \pm 0.024}$ | $\mathbf{0.253 \pm 0.022}$ | $0.261 \pm 0.020$ | $0.265 \pm 0.019$ |
|  | 336 | $\mathbf{0.267 \pm 0.032}$ | $0.268 \pm 0.030$ | $0.277 \pm 0.028$ | $0.281 \pm 0.027$ |
|  | 720 | $0.302 \pm 0.030$ | $0.306 \pm 0.029$ | $0.310 \pm 0.028$ | $\mathbf{0.301 \pm 0.029}$ |
| SolarH | 96 | $\mathbf{0.192 \pm 0.023}$ | $0.623 \pm 0.397$ | / | / |
|  | 192 | $\mathbf{0.175 \pm 0.006}$ | $0.252 \pm 0.068$ | / | / |
|  | 336 | $\mathbf{0.192 \pm 0.016}$ | $0.471 \pm 0.029$ | / | / |
|  | 720 | $\mathbf{0.216 \pm 0.034}$ | $0.465 \pm 0.063$ | / | $0.550 \pm 0.187$ |
| Traffic | 96 | $\mathbf{0.215 \pm 0.029}$ | $0.329 \pm 0.039$ | / | / |
|  | 192 | $\mathbf{0.208 \pm 0.019}$ | $0.310 \pm 0.033$ | $0.312 \pm 0.032$ | / |
|  | 336 | $\mathbf{0.237 \pm 0.028}$ | $0.307 \pm 0.038$ | / | / |
|  | 720 | $\mathbf{0.263 \pm 0.038}$ | $0.320 \pm 0.040$ | / | / |

Table 8: MAE error on the forecasting task using 3 inner-steps during training and a varying number of adaptation gradient steps at inference. Best results are in bold.

|  | H | 1 | 3 | 10 | 50 |
|---|---|---|---|---|---|
| Electricity | 96 | $0.259 \pm 0.020$ | $\mathbf{0.222 \pm 0.018}$ | $\mathbf{0.222 \pm 0.017}$ | $0.228 \pm 0.019$ |
|  | 192 | $0.269 \pm 0.020$ | $\mathbf{0.230 \pm 0.026}$ | $0.232 \pm 0.020$ | $0.233 \pm 0.026$ |
|  | 336 | $0.273 \pm 0.033$ | $\mathbf{0.262 \pm 0.031}$ | $0.264 \pm 0.032$ | $0.268 \pm 0.032$ |
|  | 720 | $0.351 \pm 0.038$ | $0.303 \pm 0.041$ | $0.300 \pm 0.040$ | $\mathbf{0.299 \pm 0.039}$ |
| SolarH | 96 | $0.487 \pm 0.196$ | $\mathbf{0.179 \pm 0.003}$ | $0.181 \pm 0.003$ | $0.186 \pm 0.003$ |
|  | 192 | $0.411 \pm 0.088$ | $\mathbf{0.193 \pm 0.015}$ | $0.195 \pm 0.014$ | $0.199 \pm 0.013$ |
|  | 336 | $0.435 \pm 0.153$ | $\mathbf{0.189 \pm 0.013}$ | $0.203 \pm 0.006$ | $0.223 \pm 0.012$ |
|  | 720 | $0.394 \pm 0.173$ | $0.209 \pm 0.029$ | $\mathbf{0.203 \pm 0.006}$ | $0.209 \pm 0.027$ |
| Traffic | 96 | $0.320 \pm 0.038$ | $\mathbf{0.215 \pm 0.037}$ | $0.219 \pm 0.043$ | $0.226 \pm 0.046$ |
|  | 192 | $0.299 \pm 0.023$ | $\mathbf{0.206 \pm 0.023}$ | $0.209 \pm 0.026$ | $0.214 \pm 0.027$ |
|  | 336 | $0.345 \pm 0.038$ | $\mathbf{0.226 \pm 0.030}$ | $0.228 \pm 0.031$ | $0.233 \pm 0.032$ |
|  | 720 | $0.321 \pm 0.034$ | $\mathbf{0.259 \pm 0.038}$ | $0.260 \pm 0.038$ | $0.266 \pm 0.039$ |

Table 9: MAE error on the forecasting task using 10 inner-steps during training and a varying number of adaptation gradient steps at inference. Best results are in bold.

|  | H | 1 | 3 | 10 | 50 |
|---|---|---|---|---|---|
| Electricity | 96 | $0.381 \pm 0.030$ | $0.249 \pm 0.024$ | $\mathbf{0.236 \pm 0.024}$ | $0.238 \pm 0.024$ |
|  | 192 | $0.448 \pm 0.045$ | $0.273 \pm 0.019$ | $0.244 \pm 0.014$ | $\mathbf{0.244 \pm 0.013}$ |
|  | 336 | $0.514 \pm 0.053$ | $0.283 \pm 0.033$ | $\mathbf{0.241 \pm 0.025}$ | $0.242 \pm 0.024$ |
|  | 720 | $0.647 \pm 0.068$ | $0.400 \pm 0.051$ | $\mathbf{0.286 \pm 0.023}$ | $0.287 \pm 0.021$ |
| SolarH | 96 | $0.605 \pm 0.029$ | $0.380 \pm 0.018$ | $\mathbf{0.188 \pm 0.012}$ | $0.199 \pm 0.015$ |
|  | 192 | $0.382 \pm 0.072$ | $0.250 \pm 0.012$ | $\mathbf{0.202 \pm 0.034}$ | $0.204 \pm 0.035$ |
|  | 336 | $0.745 \pm 0.105$ | $0.431 \pm 0.221$ | $\mathbf{0.201 \pm 0.033}$ | $0.208 \pm 0.032$ |
|  | 720 | $0.745 \pm 0.082$ | $0.477 \pm 0.039$ | $0.205 \pm 0.030$ | $\mathbf{0.205 \pm 0.029}$ |
| Traffic | 96 | $0.450 \pm 0.023$ | $0.273 \pm 0.026$ | $\mathbf{0.225 \pm 0.028}$ | $0.230 \pm 0.034$ |
|  | 192 | $0.506 \pm 0.028$ | $0.318 \pm 0.021$ | $\mathbf{0.233 \pm 0.022}$ | $0.236 \pm 0.026$ |
|  | 336 | $0.500 \pm 0.042$ | $0.320 \pm 0.021$ | $\mathbf{0.247 \pm 0.028}$ | $0.249 \pm 0.031$ |
|  | 720 | $0.511 \pm 0.035$ | $0.323 \pm 0.022$ | $\mathbf{0.266 \pm 0.027}$ | $0.272 \pm 0.024$ |

**Results** We conduct more extensive experiments in Table 7, Table 8, Table 9 to quantify the MAE score variation according to different number of gradient steps during training and inference. The tables show that using the same number of steps in training and inference leads to better results. This is expected since using different gradient steps for training and inference makes the inference model slightly different from the training model. In addition, using 3 gradient steps instead of 1 clearly improves the performances, but using 10 instead of 3 does not. Indeed, it usually leads to overall better results for longer horizon, but the gain is not clear for smaller horizons. Hence using 3 gradient steps is a suitable choice.

### A.2.4 TIMEFLOW VARIANTS WITH OTHER META-LEARNING TECHNIQUES

**Baselines** Before converging to the current architecture and optimization of TimeFlow, we explored different options to condition the INR with the observations. The first one was inspired by the neural process architecture, which uses a set encoder to transform a set of observations $(t_i, x_{t_i})_{i \in \mathcal{I}}$ into a latent code $z$ by applying a pooling layer after a feed forward network. We observed that this encoder in combination with the modulated fourier features network was able to achieve relatively good results on the forecasting task but suffered of underfitting on more complex datasets such as *Electricity*.

This led us to consider auto-decoding methods instead, i.e. encoder-less architectures for conditioning the weights of the coordinate-based network. We trained TimeFlow with the REPTILE algorithm (Nichol et al., 2018), which is a first-order meta-learning technique that adapts the code in a few steps of gradient descent. In contrast with a second-order method, we observed that REPTILE was less costly to train but struggled to escape sub optimal minima, which led to unstable training and underfitting.

From an implementation point of view, the only difference between second order and first order, is that in the latter the code is detached from the computation graph before taking the outer-loop parameter update. When the code is not detached, it remains a function of the common parameters $z = z(\theta, w)$, which means that the computation graph for the outer-loop also includes the inner-loop updates to the codes. Therefore the outer-loop gradient update involves a gradient through a gradient

and requires an additional backward pass through the INR to compute the Hessian. Please refer to Finn et al. (2017) for more technical details.

Table 10: Comparison of second-order and first-order (REPTILE) meta learning for TimeFlow on the imputation task. Mean MAE results on the missing grid over five different time windows. $\tau$ stands for the subsampling rate. Bold results are best.

|  | $\tau$ | TimeFlow | TimeFlow w REPTILE |
|---|---|---|---|
| Electricity | 0.05 | **0.324 ± 0.013** | 0.363 ± 0.062 |
|  | 0.10 | **0.250 ± 0.010** | 0.343 ± 0.036 |
|  | 0.20 | **0.225 ± 0.008** | 0.312 ± 0.043 |
|  | 0.30 | **0.212 ± 0.007** | 0.308 ± 0.035 |
|  | 0.50 | **0.194 ± 0.007** | 0.305 ± 0.046 |
| Solar | 0.05 | **0.095 ± 0.015** | 0.125 ± 0.025 |
|  | 0.10 | **0.083 ± 0.015** | 0.123 ± 0.032 |
|  | 0.20 | **0.072 ± 0.015** | 0.108 ± 0.021 |
|  | 0.30 | **0.061 ± 0.012** | 0.105 ± 0.027 |
|  | 0.50 | **0.054 ± 0.013** | 0.102 ± 0.021 |
| Traffic | 0.05 | **0.283 ± 0.016** | 0.304 ± 0.026 |
|  | 0.10 | **0.211 ± 0.012** | 0.264 ± 0.009 |
|  | 0.20 | **0.168 ± 0.006** | 0.242 ± 0.019 |
|  | 0.30 | **0.151 ± 0.007** | 0.218 ± 0.020 |
|  | 0.50 | **0.139 ± 0.007** | 0.216 ± 0.017 |

**Results** In Table 10, we show the performance of first-order TimeFlow on the imputation task. In low sampling regimes the difference with TimeFlow is less perceptive, but its performance plateaus when the number of points increases. This is not surprising. Indeed, as though the task is actually simpler when $\tau$ increases, the optimization is made more difficult with the increased number of observations. We provide the performance of TimeFlow with a set encoder on the Forecasting task in Table 11. We observed that this version failed to generalize well for complex datasets.

Table 11: Comparison of optimization-based and set-encoder-based meta learning for TimeFlow on the forecasting task. Mean MAE forecast results over different time windows. $H$ stands for the horizon. Bold results are best.

|  | $H$ | TimeFlow | TimeFlow w set encoder |
|---|---|---|---|
| Electricity | 96 | **0.228 ± 0.026** | 0.362 ± 0.032 |
|  | 192 | **0.238 ± 0.020** | 0.360 ± 0.028 |
|  | 336 | **0.270 ± 0.031** | 0.382 ± 0.038 |
|  | 720 | **0.316 ± 0.055** | 0.431 ± 0.059 |
| SolarH | 96 | **0.190 ± 0.013** | 0.251 ± 0.071 |
|  | 192 | **0.202 ± 0.020** | 0.239 ± 0.058 |
|  | 336 | **0.209 ± 0.017** | 0.235 ± 0.040 |
|  | 720 | **0.218 ± 0.048** | 0.231 ± 0.032 |
| Traffic | 96 | **0.217 ± 0.036** | 0.276 ± 0.031 |
|  | 192 | **0.212 ± 0.028** | 0.281 ± 0.034 |
|  | 336 | **0.238 ± 0.034** | 0.297 ± 0.042 |
|  | 720 | **0.279 ± 0.050** | 0.333 ± 0.048 |

### A.2.5 INFLUENCE OF THE MODULATION

In TimeFlow, we apply shift modulations to the parameters of the INR, i.e. for each layer $l$ we only modify the biases of the network with an extra bias term $\phi_l^{(j)}$. We generate these bias terms with a linear hypernetwork that maps the code $z^{(j)}$ to the modulations. The output of the $l$-th layer of the modulated INR is thus given by $\phi_{l+1} = \text{ReLU}(\theta_l \phi_{l-1} + b_l + \psi_l^{(j)})$, where $\psi_l^{(j)} = W_l z^{(j)}$ and $(W_l)_{l=1}^L$ are parameters of the hypernetwork. However, another common modulation is the combination of the scale and shift modulation, which leads to the output of the $l$-th layer of the modulated INR being given by $\phi_{l+1} = \text{ReLU}((S_l z^{(j)}) \circ (\theta_l \phi_{l-1} + b_l) + \psi_l^{(j)})$, where $\psi_l^{(j)} = W_l z^{(j)}$, and $(W_l)_{l=1}^L$ and $(S_l)_{l=1}^L$ are parameters of the hypernetwork and $\circ$ is the Hadamard product.

In Table 12, we conduct additional experiments on the *Electricity* dataset in the forecasting setting with different time horizons. In these experiments, we compare two scenarios: one where the INR is modulated only by a shift factor and the other where the INR is modulated by both a shift and a scale factor. We kept the architecture and hyperparameters consistent with those described in

Appendix A.1. The experiments shown in Table 12 indicate that the INR is longer to train with shift and scale modulations due to the increased number of parameters involved. Furthermore, we observe that the shift and scale modulated INR performed similarly or even worse than the INR with only shift modulation. These two drawbacks, namely an increased computational time and similar or worse performances, motivate modulating the INR only by a shift factor.

Table 12: Ablation on modulations for the forecasting task on *Electricity* dataset for different horizons. Models are trained on a given time window and tested on four new time windows. Models are trained on a single NVIDIA TITAN RTX GPU.

|  | 96 | | 192 | | 336 | | 720 | |
|---|---|---|---|---|---|---|---|---|
|  | MAE | Training time | MAE | Training time | MAE | Training time | MAE | Training time |
| Shift | $0.233 \pm 0.014$ | 2h30 | $0.245 \pm 0.016$ | 2h31 | $0.264 \pm 0.020$ | 2h33 | $0.303 \pm 0.041$ | 2h46 |
| Shift and scale | $0.257 \pm 0.019$ | 3h29 | $0.263 \pm 0.014$ | 3h32 | $0.268 \pm 0.025$ | 3h45 | $0.308 \pm 0.037$ | 4h14 |

### A.2.6 DISCUSSION ON OTHER HYPERPARAMETERS

While the dimension of $z$ is indeed a crucial hyperparameter, it is important to note that other hyperparameters also play a significant role in the performance of the INR. For example, the number of layers in the FFN directly affects the ability of the model to fit the time series. In our experiments, we have observed that using five or more layers yields good performance, and including additional layers can lead to slight improvements in the generalization settings.

Similarly, the number of frequencies used in the frequency embedding is another important hyperparameter. Using too few frequencies can limit the network's ability to capture patterns, while using too many frequencies can hinder its ability to generalize accurately.

The choice of learning rate is critical for achieving stable convergence during training. Therefore, in practice, we use a low learning rate combined with a cosine annealing scheduler to ensure stable and effective training.

## B DATASETS, SCORES AND NORMALIZATION

For the complete datasets, *Electricity* dataset is available here, *Traffic* dataset here and *Solar* data set here.

**Datasets information**  The *Electricity* dataset comprises hourly electricity load curves of 321 customers in Portugal, spanning the years 2012 to 2014. The *Traffic* dataset is composed of hourly road occupancy rates from 862 locations in San Francisco during 2015 and 2016. Lastly, the *Solar* dataset contains measurements of solar power production from 137 photovoltaic plants in Alabama, recorded at 10-minute intervals in 2006. Additionally, we have created an hourly version, *SolarH*, for the sake of consistency in the forecasting section. These datasets exhibit diversity in various characteristics: • They exhibit diverse temporal frequencies, including daily and weekly seasonality observed in the *Traffic* and *Electricity* datasets, while the *Solar* dataset possesses only daily frequency. • There is individual variability across data samples and more pronounced trends in the *Electricity* dataset compared to the *Traffic* and *Solar* datasets. Table 13 provides a concise overview of the main information about the datasets used for forecasting and imputation tasks.

Table 13: Summary of datasets information

| Dataset name | Number of samples | Number of time steps | Sampling frequency | Location | Years |
|---|---|---|---|---|---|
| Electricity | 321 | 26 304 | hourly | Portugal | $2012 - 2014$ |
| Traffic | 862 | 17 544 | hourly | San Francisco bay | $2015 - 2016$ |
| Solar | 137 | 52 560 | 10 minutes | Alabama | 2006 |
| SolarH | 137 | 8 760 | hourly | Alabama | 2006 |

**How TimeFlow relative improvement score is computed**   In many paper tables, the TimeFlow improvement score appears in the last row of the table. Its purpose is to quantify the average marginal gain of TimeFlow over the method under consideration. It is computed as follows:

$$\text{TimeFlow improvement} = \frac{1}{L} \sum_{l=1}^{L} \frac{s_l(\text{baseline}) - s_l(\text{TimeFlow})}{s_l(\text{baseline})}$$

- s stands for the Mean Absolute Error score of the considered method against the ground truth at line l (for instance in Table 1, $s_1(\text{TimeFlow})= 0.324$, $s_2(\text{TimeFlow})= 0.250$ etc).
- L stands for the number of line in the table (for instance 15 in Table 1).

**z-normalization**   To preprocess each dataset, we apply the widely used z-normalization technique per-sample $j$ on the entire series: $x_{norm}^{(j)} = \frac{x^{(j)} - \text{mean}(x^{(j)})}{\text{std}(x^{(j)})}$.

# C   IMPUTATION EXPERIMENTS

## C.1   MODELS COMPLEXITY

We can see in Table 14 that our method has fewer parameters than SOTA imputation methods, 10 times less than BRITS and 20 times less than SAITS. It is mainly due to their modelisation of interaction between samples. SAITS, which is based on transformers has the highest number of parameters when mTAN has the lowest number of parameters.

Table 14: Number of parameters for each DL methods on the imputation task on the *Electricity* dataset.

|  | TimeFlow | DeepTime | NeuralProcess | mTAN | SAITS | BRITS | TIDER |
|---|---|---|---|---|---|---|---|
| Number of parameters | 602k | 1315k | 248k | 113k | 11 137k | 6 220k | 1 034k |

## C.2   IMPUTATION FOR PREVIOUSLY UNSEEN TIME SERIES

**Setting**   In this section we analyze in details the imputations results for previously unseen time series described in Section 4.1. Specifically, TimeFlow is trained on a given set of time series within a defined time window and then used for inference on new time series. We train TimeFlow on 50 % of the samples and consider the remaining 50 % as the new time series.

We compare in Table 15 observed grid fit scores and missing grid inference scores for time series known at training and time series unknown at training.

Table 15: TimeFlow MAE imputation errors results for imputation previsouly unseen time series.

| | $\tau$ | Known time series Fit | Known time series Inference | New time series Fit | New time series Inference |
|---|---|---|---|---|---|
| | 0.05 | $0.060 \pm 0.010$ | $0.402 \pm 0.021$ | $0.142 \pm 0.083$ | $0.413 \pm 0.026$ |
| | 0.10 | $0.046 \pm 0.006$ | $0.302 \pm 0.010$ | $0.144 \pm 0.098$ | $0.309 \pm 0.016$ |
| Electricity | 0.20 | $0.067 \pm 0.015$ | $0.285 \pm 0.014$ | $0.154 \pm 0.089$ | $0.291 \pm 0.022$ |
| | 0.30 | $0.093 \pm 0.022$ | $0.266 \pm 0.010$ | $0.163 \pm 0.073$ | $0.271 \pm 0.017$ |
| | 0.50 | $0.108 \pm 0.012$ | $0.236 \pm 0.010$ | $0.167 \pm 0.061$ | $0.245 \pm 0.017$ |
| | 0.05 | $0.014 \pm 0.002$ | $0.104 \pm 0.015$ | $0.050 \pm 0.037$ | $0.109 \pm 0.016$ |
| | 0.10 | $0.017 \pm 0.002$ | $0.092 \pm 0.015$ | $0.052 \pm 0.036$ | $0.099 \pm 0.017$ |
| Solar | 0.20 | $0.028 \pm 0.008$ | $0.078 \pm 0.014$ | $0.058 \pm 0.031$ | $0.089 \pm 0.017$ |
| | 0.30 | $0.038 \pm 0.009$ | $0.072 \pm 0.013$ | $0.063 \pm 0.028$ | $0.084 \pm 0.018$ |
| | 0.50 | $0.045 \pm 0.011$ | $0.066 \pm 0.013$ | $0.067 \pm 0.025$ | $0.080 \pm 0.019$ |
| | 0.05 | $0.044 \pm 0.003$ | $0.291 \pm 0.013$ | $094 \pm 0.051$ | $0.291 \pm 0.012$ |
| | 0.10 | $0.033 \pm 0.001$ | $0.209 \pm 0.010$ | $0.093 \pm 0.060$ | $0.216 \pm 0.012$ |
| Traffic | 0.20 | $0.037 \pm 0.006$ | $0.175 \pm 0.008$ | $0.095 \pm 0.058$ | $0.186 \pm 0.013$ |
| | 0.30 | $0.048 \pm 0.005$ | $0.164 \pm 0.006$ | $0.098 \pm 0.051$ | $0.175 \pm 0.013$ |
| | 0.50 | $0.068 \pm 0.004$ | $0.159 \pm 0.007$ | $0.110 \pm 0.042$ | $0.169 \pm 0.012$ |

**Results** The results presented in Table 15 indicate that the inference MAE for missing grids shows consistency between known and new samples, regardless of the data or sampling rate. However, it is worth noting that there is a slight drop in performance compared to the results in table Table 1. This decrease is because in Table 15, the shared architecture is trained on only half the samples, affecting its overall performance.

## C.3 DETAILS ON DEEPTIME ADAPTATION FOR IMPUTATION

As DeepTime was proposed to address the forecasting task with a deeptime-index model, the authors did not tackle the task of imputation and left it out for future work. Given the success of this method and the motivation of our work, we wanted to explore its capabilities to impute time series with several subsampling rates. Following our current framework, we first tried to train the model in a self-supervised way, i.e. trying to reconstruct observations $x^{(j)} \in \mathcal{T}^{(j)}$ after the INR has been conditioned with the Ridge Regressor on the same set of observations, but discovered failure cases for $\tau \leq 0.20$. To be faithful to the original supervised training of DeepTime, we therefore randomly mask out 50% of the observations that we use as context for the Ridge Regressor and try to infer the other 50% (the targets) to train the INR.

We provide a qualitative comparison of the model's performance with these two different training procedures in Figure 6. We can notice that the model that results from the self-supervised training perfectly fits the observations but completely misses the important patterns of the series. On the other hand, when DeepTime is trained to infer target values based on observations, it is able to capture the general trends. We think that in the small subsampling regime ($\tau \leq 0.20$), the Ridge Regressor easily fits very well all the observations which hinders the training of the INR's basis.

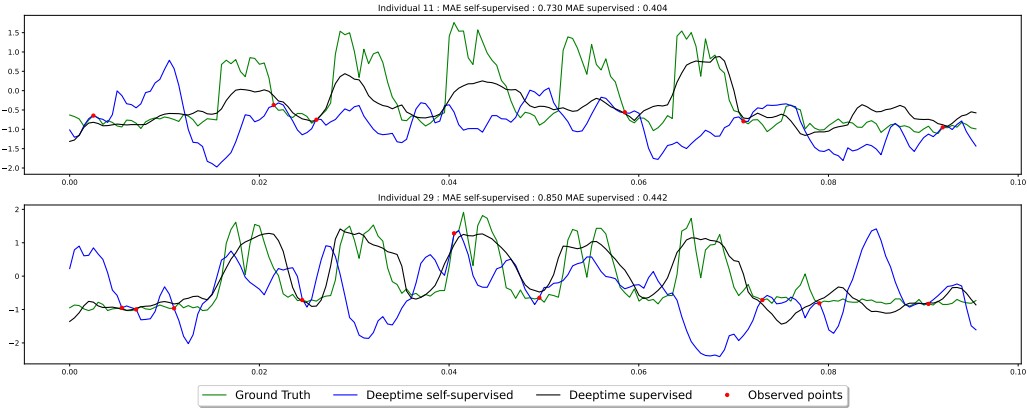

Figure 6: *Electricity dataset*. Self supervised DeepTime imputation (blue line) and supervised Deep-Time imputation (black line) with 5% of known point (red points) on the eight first days of samples 11 (top) and 29 (bottom).

## C.4 IMPUTATION AGAINST NON DEEP LEARNING METHODS

**Setting** In addition to the deep learning methods presented in Table 1, we evalute TimeFlow against two classic machine learning baselines, K-Nearest Neighbours (KNN) and linear interpolation, which are valuable for getting an idea of the complexity of the problem.

Table 16: Mean MAE imputation results on the missing grid only over five different time window. $\tau$ stands for the subsampling rate. Bold results are best, underline results are second best.

|  | $\tau$ | TimeFlow | Linear interpolation | KNN (k=3) |
|---|---|---|---|---|
| | 0.05 | **0.324 ± 0.013** | 0.828 ± 0.045 | 0.531 ± 0.033 |
| | 0.10 | **0.250 ± 0.010** | 0.716 ± 0.039 | 0.416 ± 0.020 |
| Electricity | 0.20 | **0.225 ± 0.008** | 0.518 ± 0.029 | 0.363 ± 0.019 |
| | 0.30 | **0.212 ± 0.007** | 0.396 ± 0.022 | 0.342 ± 0.017 |
| | 0.50 | **0.194 ± 0.007** | 0.275 ± 0.015 | 0.323 ± 0.016 |
| | 0.05 | **0.095 ± 0.015** | 0.339 ± 0.031 | 0.151 ± 0.017 |
| | 0.10 | **0.083 ± 0.015** | 0.170 ± 0.014 | 0.128 ± 0.017 |
| Solar | 0.20 | **0.072 ± 0.015** | 0.088 ± 0.010 | 0.110 ± 0.016 |
| | 0.30 | **0.061 ± 0.012** | 0.063 ± 0.009 | 0.103 ± 0.017 |
| | 0.50 | 0.054 ± 0.013 | **0.044 ± 0.008** | 0.096 ± 0.016 |
| | 0.05 | **0.283 ± 0.016** | 0.813 ± 0.027 | 0.387 ± 0.014 |
| | 0.10 | **0.211 ± 0.012** | 0.701 ± 0.026 | 0.293 ± 0.012 |
| Traffic | 0.20 | **0.168 ± 0.006** | 0.508 ± 0.022 | 0.249 ± 0.010 |
| | 0.30 | **0.151 ± 0.007** | 0.387 ± 0.018 | 0.228 ± 0.009 |
| | 0.50 | **0.139 ± 0.007** | 0.263 ± 0.013 | 0.204 ± 0.009 |
| TimeFlow improvement | / | | 49.06 % | 35.95 % |

**Results** KNN imputation uses information from other individuals and gives satisfactory results at all sampling rates. On the other hand, the purely univariate approach of linear interpolation struggles at low sampling rates but performs well at high sampling rates. TimeFlow significantly outperforms both baselines by a large margin.

## D  FORECASTING EXPERIMENTS

### D.1  DISTINCTION BETWEEN ADJACENT TIME WINDOWS AND NEW TIME WINDOWS DURING INFERENCE

In Section 4.2, we presented the forecasting results for periods outside the training period. These periods can be classified into two types: adjacent to or disjoint from the training period. Figure 7 illustrates these distinct test periods for the *Electricity* dataset. The same principle applies to the *Traffic* and *SolarH* datasets, with one notable difference: the number of test periods is smaller in these datasets compared to *Electricity* dataset due to the fewer time steps available.

In Table 2, we presented the results indistinctly for the two types of test periods: adjacent to and disjoint from the training window. Here, we aim to differentiate the results for these two types of window and emphasize their significant impact on Informer and AutoFormer results. Specifically, Table 17 showcases the results for the test periods adjacent to the training window. In contrast, Table 18 displays the results for the test periods disjointed from the training window

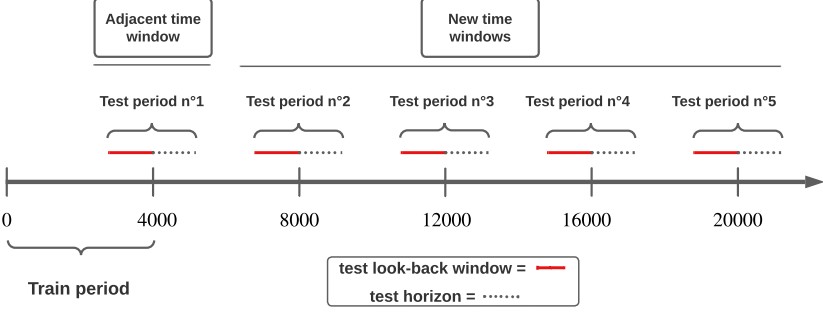

Figure 7: Distinction between adjacent time windows and new time windows during inference for the *Electricity* dataset

**Results** TimeFlow, PatchTST, DLinear and DeepTime maintain consistent forecasting results whether tested on the period adjacent to the training period or on a disjoint period. However, Auto-Former and Informer show a significant drop in performance when tested on new disjoint periods.

Table 17: Mean MAE forecast results for adjacent time windows. H stands for the horizon. Bold results are best, underline results are second best.

| | $H$ | Continuous methods | | | Discrete methods | | | |
| | | TimeFlow | DeepTime | Neural Process | Patch-TST | DLinear | AutoFormer | Informer |
|---|---|---|---|---|---|---|---|---|
| Electricity | 96 | $\underline{0.218 \pm 0.017}$ | $0.240 \pm 0.027$ | $0.392 \pm 0.045$ | $\mathbf{0.214 \pm 0.020}$ | $0.236 \pm 0.035$ | $0.310 \pm 0.031$ | $0.293 \pm 0.0184$ |
| | 192 | $\underline{0.238 \pm 0.012}$ | $0.251 \pm 0.023$ | $0.401 \pm 0.046$ | $\mathbf{0.225 \pm 0.017}$ | $0.248 \pm 0.032$ | $0.322 \pm 0.046$ | $0.336 \pm 0.032$ |
| | 336 | $\underline{0.265 \pm 0.036}$ | $0.290 \pm 0.034$ | $0.434 \pm 0.075$ | $\mathbf{0.242 \pm 0.024}$ | $0.284 \pm 0.043$ | $0.330 \pm 0.019$ | $0.405 \pm 0.044$ |
| | 720 | $\underline{0.318 \pm 0.073}$ | $0.356 \pm 0.060$ | $0.605 \pm 0.149$ | $\mathbf{0.291 \pm 0.040}$ | $0.370 \pm 0.086$ | $0.456 \pm 0.052$ | $0.489 \pm 0.072$ |
| SolarH | 96 | $\mathbf{0.172 \pm 0.017}$ | $\underline{0.197 \pm 0.002}$ | $0.221 \pm 0.048$ | $0.232 \pm 0.008$ | $0.204 \pm 0.002$ | $0.261 \pm 0.053$ | $0.273 \pm 0.023$ |
| | 192 | $\mathbf{0.198 \pm 0.010}$ | $\underline{0.202 \pm 0.014}$ | $0.244 \pm 0.048$ | $0.231 \pm 0.027$ | $0.211 \pm 0.012$ | $0.312 \pm 0.085$ | $0.256 \pm 0.026$ |
| | 336 | $\underline{0.207 \pm 0.019}$ | $\mathbf{0.200 \pm 0.012}$ | $0.241 \pm 0.005$ | $0.254 \pm 0.048$ | $0.212 \pm 0.019$ | $0.341 \pm 0.107$ | $0.287 \pm 0.006$ |
| | 720 | $\mathbf{0.215 \pm 0.016}$ | $\underline{0.240 \pm 0.011}$ | $0.403 \pm 0.147$ | $0.271 \pm 0.036$ | $0.246 \pm 0.015$ | $0.368 \pm 0.006$ | $0.341 \pm 0.049$ |
| Traffic | 96 | $\underline{0.216 \pm 0.033}$ | $0.229 \pm 0.032$ | $0.283 \pm 0.028$ | $\mathbf{0.201 \pm 0.031}$ | $0.225 \pm 0.034$ | $0.299 \pm 0.080$ | $0.324 \pm 0.113$ |
| | 192 | $\underline{0.208 \pm 0.021}$ | $0.220 \pm 0.020$ | $0.292 \pm 0.023$ | $\mathbf{0.195 \pm 0.024}$ | $0.215 \pm 0.022$ | $0.320 \pm 0.036$ | $0.321 \pm 0.052$ |
| | 336 | $\underline{0.237 \pm 0.040}$ | $0.247 \pm 0.033$ | $0.305 \pm 0.039$ | $\mathbf{0.220 \pm 0.036}$ | $0.244 \pm 0.035$ | $0.450 \pm 0.127$ | $0.394 \pm 0.066$ |
| | 720 | $\mathbf{0.266 \pm 0.048}$ | $0.290 \pm 0.045$ | $0.339 \pm 0.037$ | $\underline{0.268 \pm 0.050}$ | $0.290 \pm 0.047$ | $0.630 \pm 0.043$ | $0.441 \pm 0.055$ |
| TimeFlow improvement | | / | 6.56 % | 30.79 % | 2.64 % | 7.30 % | 35.43 % | 33.07 % |

Table 18: Mean MAE forecast results for new time windows. H stands for the horizon. Bold results are best, underline results are second best.

| | $H$ | Continuous methods | | | Discrete methods | | | |
| | | TimeFlow | DeepTime | Neural Process | Patch-TST | DLinear | AutoFormer | Informer |
|---|---|---|---|---|---|---|---|---|
| Electricity | 96 | $\underline{0.230 \pm 0.012}$ | $0.245 \pm 0.026$ | $0.392 \pm 0.045$ | $\mathbf{0.222 \pm 0.023}$ | $0.240 \pm 0.025$ | $0.606 \pm 0.281$ | $0.605 \pm 0.227$ |
| | 192 | $\underline{0.246 \pm 0.025}$ | $0.252 \pm 0.018$ | $0.401 \pm 0.046$ | $\mathbf{0.231 \pm 0.020}$ | $0.257 \pm 0.027$ | $0.545 \pm 0.186$ | $0.776 \pm 0.257$ |
| | 336 | $\underline{0.271 \pm 0.029}$ | $0.285 \pm 0.034$ | $0.434 \pm 0.076$ | $\mathbf{0.253 \pm 0.027}$ | $0.298 \pm 0.051$ | $0.571 \pm 0.181$ | $0.823 \pm 0.241$ |
| | 720 | $\underline{0.316 \pm 0.051}$ | $0.359 \pm 0.048$ | $0.607 \pm 0.15$ | $\mathbf{0.299 \pm 0.038}$ | $0.373 \pm 0.075$ | $0.674 \pm 0.245$ | $0.811 \pm 0.257$ |
| SolarH | 96 | $\underline{0.208 \pm 0.005}$ | $\mathbf{0.206 \pm 0.026}$ | $0.221 \pm 0.048$ | $0.293 \pm 0.089$ | $0.212 \pm 0.019$ | $0.228 \pm 0.027$ | $0.234 \pm 0.011$ |
| | 192 | $\mathbf{0.206 \pm 0.012}$ | $\underline{0.207 \pm 0.037}$ | $0.244 \pm 0.048$ | $0.274 \pm 0.060$ | $0.223 \pm 0.029$ | $0.356 \pm 0.122$ | $0.280 \pm 0.033$ |
| | 336 | $\underline{0.211 \pm 0.005}$ | $\mathbf{0.199 \pm 0.035}$ | $0.240 \pm 0.006$ | $0.264 \pm 0.088$ | $0.223 \pm 0.032$ | $0.327 \pm 0.029$ | $0.366 \pm 0.039$ |
| | 720 | $\mathbf{0.222 \pm 0.020}$ | $\underline{0.217 \pm 0.028}$ | $0.403 \pm 0.147$ | $0.262 \pm 0.083$ | $0.251 \pm 0.047$ | $0.335 \pm 0.075$ | $0.333 \pm 0.012$ |
| Traffic | 96 | $\underline{0.218 \pm 0.042}$ | $0.229 \pm 0.032$ | $0.283, 0.0275$ | $\mathbf{0.204 \pm 0.039}$ | $0.229 \pm 0.032$ | $0.326 \pm 0.049$ | $0.388 \pm 0.055$ |
| | 192 | $\underline{0.213 \pm 0.028}$ | $0.220 \pm 0.023$ | $0.292, 0.0236$ | $\mathbf{0.198 \pm 0.031}$ | $0.223 \pm 0.023$ | $0.575 \pm 0.254$ | $0.381 \pm 0.049$ |
| | 336 | $\underline{0.239 \pm 0.035}$ | $0.244 \pm 0.040$ | $0.305, 0.0392$ | $\mathbf{0.223 \pm 0.040}$ | $0.252 \pm 0.042$ | $0.598 \pm 0.286$ | $0.448 \pm 0.055$ |
| | 720 | $\underline{0.280 \pm 0.047}$ | $0.290 \pm 0.055$ | $0.339, 0.0375$ | $\mathbf{0.270 \pm 0.059}$ | $0.304 \pm 0.061$ | $0.641 \pm 0.072$ | $0.468 \pm 0.064$ |
| TimeFlow improvement | | / | 2.50 % | 27.75 % | 3.41 % | 6.80 % | 46.26 % | 45.53 % |

## D.2 PLOTS COMPARISON: TIMEFLOW VS PATCHTST

Table 2 demonstrates the similar forecasting performance of TimeFlow and PatchTST across all horizons. To visually represent their predictions, the figures below showcase the forecasted outcomes of these methods for two samples (24 and 38) and two horizons (96 and 192) on the *Electricity*, *SolarH*, and *Traffic* datasets.

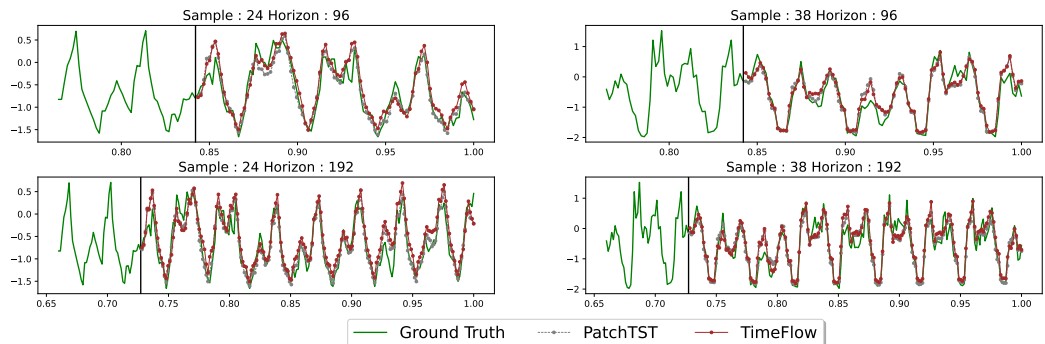

Figure 8: Qualitative comparisons of TimeFlow vs PatchTST on the *Electricity* dataset for new time windows

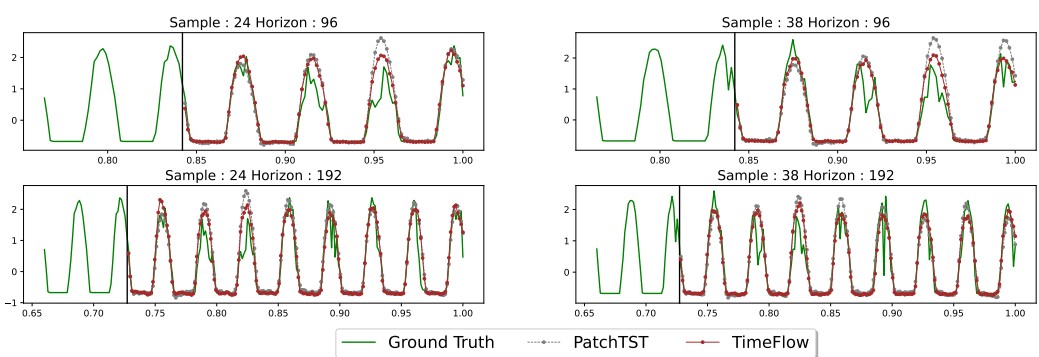

Figure 9: Qualitative comparisons of TimeFlow vs PatchTST on the *SolarH* dataset for new time windows

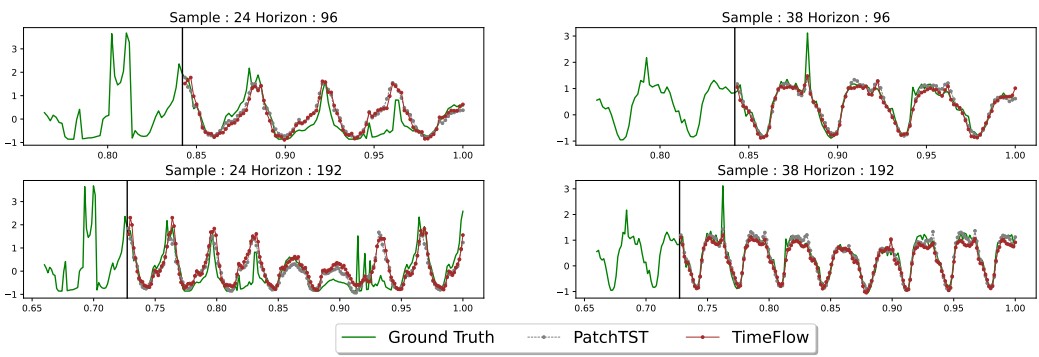

Figure 10: Qualitative comparisons of TimeFlow vs PatchTST on the *Traffic* dataset for new time windows

**Results** The visual analysis of the figures above reveals that the predictions of TimeFlow and PatchTST are remarkably similar. For instance, when examining sample 24 and horizon 192 of the *Traffic* dataset, both forecasters exhibit similar error patterns. The only noticeable distinction emerges in the *SolarH* dataset, where PatchTST tends to overestimate certain peaks.

## D.3 MODELS COMPLEXITY

In this section, we present the parameter counts and the inference time for the main forecasting baselines. Except for TimeFlow and DeepTime, the number of parameters varies with the number of samples, the look-back window, and the horizon. Thus, we report the number of parameters for two specific configurations, including a fixed dataset, a fixed look-back window, and a fixed horizon. In Table 19, we see that for PatchTST and DLinear, the larger the horizon, the more the number of parameters increases. In Table 20, it is shown that all methods' computational time increases with the horizon, which is expected. Moreover, TimeFlow is slower than the baselines that use forward computations only. Still, on the Electricity dataset, for example, the method can infer for 321 samples a horizon of 720 values with a look-back window of 512 timestamps in less than 0.2s, which does not look prohibitive for many real-world usages. This is mainly due to the small number of gradient steps at inference.

Table 19: The number of parameters for main baselines on the forecasting task on the *Electricity* dataset for horizons 96 and 720. The look-back window size is 512.

|  | TimeFlow | DeepTime | Neural Process | Patch-TST | DLinear | Informer | Autoformer |
|---|---|---|---|---|---|---|---|
| 96 | 602k | 1 315k | 480k | 1 194k | 98k | 984k | 1 005k |
| 720 | 602k | 1 315k | 480k | 6 306k | 739k | 984k | 1 005k |

Table 20: Inference time (in seconds) for the forecasting task on the *Electricity* dataset with horizons 96 and 720 and a look-back window of length 512. The statistics are computed over 10 runs using an NVIDIA TITAN RTX GPU.

|  | TimeFlow | Patch-TST | DLinear | DeepTime | AutoFormer | Informer |
|---|---|---|---|---|---|---|
| 96 | $0.147 \pm 0.007$ | $0.016 \pm 0.002$ | $0.007 \pm 0.003$ | $0.006 \pm 0.002$ | $0.027 \pm 0.001$ | $0.0191 \pm 0.002$ |
| 720 | $0.176 \pm 0.009$ | $0.020 \pm 0.001$ | $0.009 \pm 0.001$ | $0.010 \pm 0.002$ | $0.034 \pm 0.001$ | $0.0251 \pm 0.002$ |

## D.4 FORECASTING FOR PREVISOULY UNSEEN TIME SERIES

**Setting and baseline.** As mentioned in Section 4.2, most forecasters explicitly model the dependencies between samples, which limits their ability to generalize to new time series without retraining the entire model. However, TimeFlow, PatchTST, and DeepTime have the advantage of being reusable for new samples. In Table 21, we present the results of TimeFlow and PatchTST for new periods, considering both known samples and new samples. We train TimeFlow and PatchTST on 50 % of the samples and consider the remaining 50 % as the new time series.

Table 21: MAE results over horizon for the forecasting task in the context of generalization to new time series.

|  | H | Known time series | | New time series | |
|---|---|---|---|---|---|
|  |  | TimeFlow MAE error | PatchTST MAE error | TimeFlow MAE error | PatchTST MAE error |
| Electricity | 96 | $0.228 \pm 0.023$ | $0.211 \pm 0.007$ | $0.241 \pm 0.023$ | $0.224 \pm 0.020$ |
|  | 192 | $0.244 \pm 0.022$ | $0.225 \pm 0.014$ | $0.254 \pm 0.024$ | $0.238 \pm 0.024$ |
|  | 336 | $0.269 \pm 0.036$ | $0.267 \pm 0.019$ | $0.277 \pm 0.033$ | $0.285 \pm 0.005$ |
|  | 720 | $0.331 \pm 0.058$ | $0.310 \pm 0.026$ | $0.333 \pm 0.059$ | $0.331 \pm 0.045$ |
| Traffic | 96 | $0.226 \pm 0.035$ | $0.208 \pm 0.036$ | $0.222 \pm 0.031$ | $0.203 \pm 0.037$ |
|  | 192 | $0.217 \pm 0.028$ | $0.202 \pm 0.029$ | $0.215 \pm 0.026$ | $0.199 \pm 0.030$ |
|  | 336 | $0.242 \pm 0.036$ | $0.228 \pm 0.041$ | $0.240 \pm 0.031$ | $0.224 \pm 0.036$ |
|  | 720 | $0.283 \pm 0.053$ | $0.275 \pm 0.059$ | $0.283 \pm 0.049$ | $0.272 \pm 0.055$ |
| SolarH | 96 | $0.237 \pm 0.077$ | $0.256 \pm 0.055$ | $0.236 \pm 0.081$ | $0.256 \pm 0.062$ |
|  | 192 | $0.238 \pm 0.051$ | $0.251 \pm 0.239$ | $0.239 \pm 0.058$ | $0.250 \pm 0.050$ |
|  | 336 | $0.220 \pm 0.027$ | $0.255 \pm 0.663$ | $0.220 \pm 0.034$ | $0.255 \pm 0.066$ |
|  | 720 | $0.240 \pm 0.039$ | $0.267 \pm 0.062$ | $0.240 \pm 0.042$ | $0.267 \pm 0.063$ |
| TimeFlow improvement |  | / | -1.05 % | / | -0.39 % |

**Results** Table 21 demonstrates the good adaptability of both methods to new samples, as the difference in MAE between known and new samples is marginal.

## D.5 SPARSELY OBSERVED LOOK-BACK WINDOW: COMPARISON WITH PATCH-TST

**Setting and baseline.** Let's consider a setting where at inference time, the look-back window is sparsely observed. Models such as PatchTST must proceed in two steps: (i) completing the look-back window on a dense regular grid using imputation; (ii) apply the model on the completed window to predict the future. We compared TimeFlow with the following two-step processing baseline: linear interpolation handling the missing values within the partially observed look-back window, and PatchTST handling the forecasting task. We conducted experiments on the *Traffic* and *Electricity* datasets, focusing on the 96 and 192 horizons. In Table 22, we present the results at different sampling rates $\tau \in \{0.5, 0.2, 0.1\}$ within the look-back window.

Table 22: MAE results for forecasting on new samples and new period with missing values in the look-back window. Best results are in bold.

|  | H | $\tau$ | TimeFlow | | Linear interpo + PatchTST | |
|---|---|---|---|---|---|---|
|  |  |  | Imputation error | Forecast error | Imputation error | Forecast error |
| Electricity | 96 | 1. | $0.000 \pm 0.000$ | $0.228 \pm 0.028$ | $0.000 \pm 0.000$ | $\mathbf{0.221 \pm 0.023}$ |
|  |  | 0.5 | $\mathbf{0.151 \pm 0.003}$ | $\mathbf{0.239 \pm 0.013}$ | $0.257 \pm 0.008$ | $0.279 \pm 0.026$ |
|  |  | 0.2 | $\mathbf{0.208 \pm 0.006}$ | $\mathbf{0.260 \pm 0.015}$ | $0.482 \pm 0.019$ | $0.451 \pm 0.042$ |
|  |  | 0.1 | $\mathbf{0.272 \pm 0.006}$ | $\mathbf{0.295 \pm 0.016}$ | $0.663 \pm 0.029$ | $0.634 \pm 0.053$ |
|  | 192 | 1. | $0.000 \pm 0.000$ | $0.238 \pm 0.020$ | $0.000 \pm 0.000$ | $\mathbf{0.229 \pm 0.020}$ |
|  |  | 0.5 | $\mathbf{0.149 \pm 0.004}$ | $\mathbf{0.235 \pm 0.011}$ | $0.258 \pm 0.006$ | $0.280 \pm 0.032$ |
|  |  | 0.2 | $\mathbf{0.209 \pm 0.006}$ | $\mathbf{0.257 \pm 0.013}$ | $0.481 \pm 0.021$ | $0.450 \pm 0.054$ |
|  |  | 0.1 | $\mathbf{0.274 \pm 0.010}$ | $\mathbf{0.289 \pm 0.016}$ | $0.669 \pm 0.030$ | $0.650 \pm 0.060$ |
| Traffic | 96 | 1. | $0.000 \pm 0.000$ | $0.217 \pm 0.032$ | $0.000 \pm 0.000$ | $\mathbf{0.203 \pm 0.037}$ |
|  |  | 0.5 | $\mathbf{0.219 \pm 0.017}$ | $\mathbf{0.224 \pm 0.033}$ | $0.276 \pm 0.012$ | $0.255 \pm 0.041$ |
|  |  | 0.2 | $\mathbf{0.278 \pm 0.017}$ | $\mathbf{0.252 \pm 0.029}$ | $0.532 \pm 0.017$ | $0.483 \pm 0.040$ |
|  |  | 0.1 | $\mathbf{0.418 \pm 0.019}$ | $\mathbf{0.382 \pm 0.014}$ | $0.738 \pm 0.023$ | $0.721 \pm 0.073$ |
|  | 192 | 1. | $0.000 \pm 0.000$ | $0.212 \pm 0.028$ | $0.000 \pm 0.000$ | $\mathbf{0.197 \pm 0.030}$ |
|  |  | 0.5 | $\mathbf{0.176 \pm 0.014}$ | $\mathbf{0.217 \pm 0.017}$ | $0.276 \pm 0.011$ | $0.245 \pm 0.029$ |
|  |  | 0.2 | $\mathbf{0.233 \pm 0.017}$ | $\mathbf{0.236 \pm 0.021}$ | $0.532 \pm 0.020$ | $0.480 \pm 0.050$ |
|  |  | 0.1 | $\mathbf{0.304 \pm 0.019}$ | $\mathbf{0.277 \pm 0.021}$ | $0.734 \pm 0.022$ | $0.787 \pm 0.172$ |

**Results.** Although PatchTST performs slightly better with a dense look-back window, its performance significantly deteriorates as the value of $\tau$ decreases. In contrast, the performance of TimeFlow is only minimally affected by the reduction in the sampling rate.

## D.6 INFLUENCE OF THE LOOK-BACK WINDOW FOR FORECASTING

In Figure 11, it is shown that both excessively short and overly long look-back windows can harm TimeFlow forecasting performance. More precisely, the performances increases with the look-back window size up to a certain size, where the performances then drop slowly.

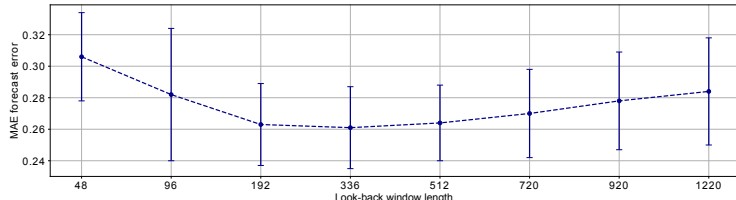

Figure 11: MAE forecast error per look-back windows length for the *Electricity* dataset (horizon window length is 336). The model is trained on a given time window and tested on four new time windows.

### D.7   INFLUENCE OF THE HORIZON LENGTH FOR FORECASTING

In Figure 12, it is shown that the performances decrease with the length of the horizon. This is to be expected, since the longer the horizon, the harder the task.

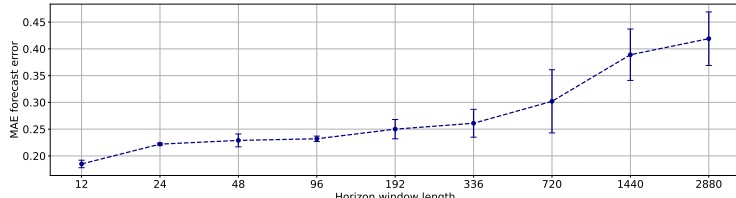

Figure 12: MAE forecast error per horizons length for the *Electricity* dataset (look-back window length is 512). The model is trained on a given time window and tested on four new time windows.

## E   LATENT SPACE EXPLORATION

### E.1   LATENT SPACE INTERPOLATION BETWEEN TWO LEARNED CODES

It is interesting to understand how the latent space behaves between two learned codes, $z_1$ and $z_2$, which are representations of $x_1$ and $x_2$. We propose in Figure 13 to visualize how new $z_\lambda = \lambda z_1 + (1 - \lambda)z_2$ are decoded in the time-series domain ($f_{\theta, h_w(z_\lambda)}(t)$ values).

**Setting.**   We choose two $z_1$ and $z_2$ learned from the *Electricity* dataset and interpolate these two latent codes for $\lambda \in \{0.0, 0.1, 0.25, 0.50, 0.75, 0.9, 1.\}$.

**Results.**   In Figure 13, we observe that the interpolation path between two codes yields a smooth transition in the time series domain. This suggests that the latent space is smooth and well-structured, and that the learned representations captured meaningful features of the time series, which could explain TimeFlow's generalization property.

### E.2   TIMEFLOW SENSITIVITY TO MODULATIONS PERTURBATION

In the preceding section, we observed the smoothness of the latent space. A crucial question arises: can we interpret each dimension of the latent space independently ?

**Setting**   In this setting, we perturbed a specific dimension of the modulation (by adding Gaussian noise) for only one particular layer of the INR. Then, we observe the difference in the time domain between the non-perturbated TimeFlow and the perturbated one. For instance in Figure 14, we add noise only for the third layer of the INR and the 50th channel.

**Results.**   In Figure 14, we observed that adding a small perturbation added a smooth daily frequency pattern. In Figure 15, we observed that adding a small perturbation induces a bias that impacts the high frequencies but does not affect the low frequencies. Finally, in Figure 16, adding a small perturbation induces a very local and slight bias (the effect is almost null). In conclusion, the impact of the small perturbation depends on the channel and the layer, but it is hard to interpret each dimension independently.

### E.3   VISUALIZATION OF TWO CODE DISTRIBUTIONS IN THE LATENT SPACE

Examining the behavior of the latent space at the instance level is of particular interest. It provides insights into how individual time series evolve within the latent space. However, exploring the latent space between two time series distributions is also crucial.

**Setting.**   We propose encoding all samples (321 samples) from the *Electricity* dataset for two distinct time periods (each period is about 25 days $\approx 600$ timestamps). This results in two distributions

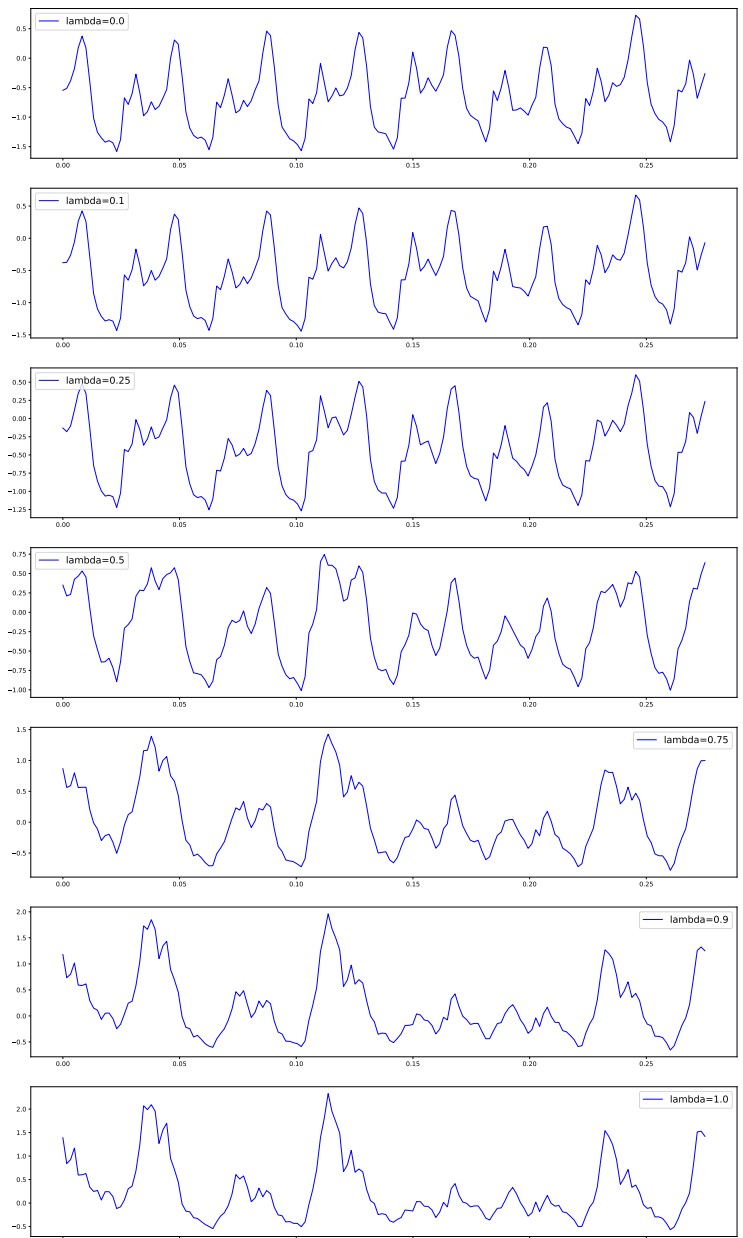

Figure 13: Visualization of the reconstructed time series for different linear interpolation of the two codes $z_1$ and $z_2$ learned from the *Electricity* dataset.

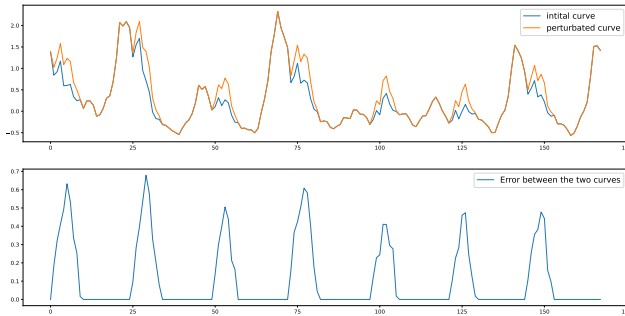

Figure 14: Effect of adding a small perturbation to the modulation shift of the third layer and 50th channel.

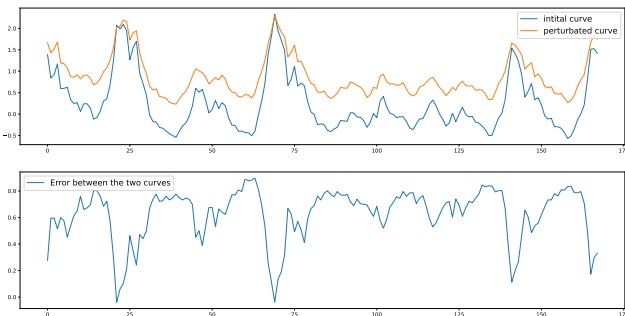

Figure 15: Effect of adding a small perturbation to the modulation shift of the third layer and 51th channel.

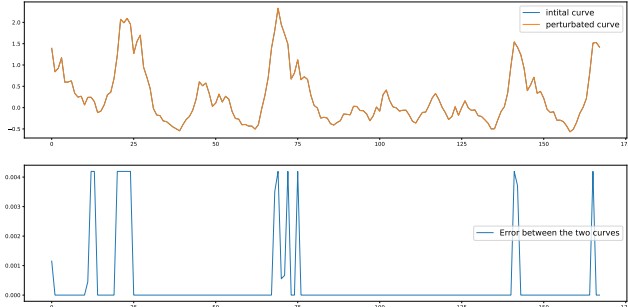

Figure 16: Effect of adding a small perturbation to the modulation shift of the fourth layer and 50th channel.

of latent codes, each representing different temporal support. Then, we employ Principal Component Analysis (PCA) to visualize these two latent code distributions in a two-dimensional space, as illustrated in Figure 17. This visualization allows us to explore the structural differences, similarities, and temporal variations in the latent space representations across the specified time intervals. In Figure 17a, the two compared time period are separated by approximately 3 months ($\approx$ 2000 timestamps). In Figure 17b, the two compared time period are separated by approximately 6 months ($\approx$ 4000 timestamps).

**Results.** Figure 17a shows that when the temporal periods are not too far from each other, the distributions of codes can largely overlap in the 2D visualization from the PCA. Conversely, as illustrated in Figure 17b, when the temporal periods are far from each other, the distributions of codes between the two periods become more distinct in the 2D visualization. This observation suggests that the proximity or disparity in temporal distribution shift influences the separability of

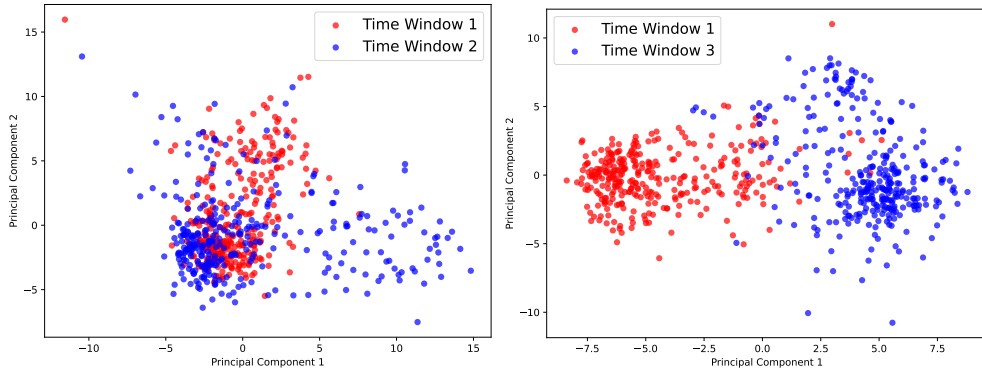

(a) The temporal shift between the two codes distribution is 3 months. (b) The temporal shift between the two codes distribution is 6 months.

Figure 17: Visualization of the two first PCA axes for two distributions of the latent codes (temporal distribution shift).

latent representations in the 2D PCA space. However, this presumed separability in the latent space does not seem to significantly impact the generalization performance of TimeFlow across time, as evidenced by the results presented in Table 2 and Table 18. This suggests that our INR can handle relatively diverse code distributions.

