# OpenReview forum: "Time Series Continuous Modeling for Imputation and Forecasting with Implicit Neural Representations"
_ICLR.cc/2024/Conference — Submitted to ICLR 2024_

### Official Review · Reviewer_TAfq · 2023-10-30

**Soundness:** 4 excellent
**Presentation:** 4 excellent
**Contribution:** 3 good
**Rating:** 8
**Confidence:** 3

**Summary:**

This paper presents a new method, TimeFlow, for time series analysis aimed to address imputation and forecasting tasks under the realistic issues of irregularly sampled and unaligned data. The authors compare with many SOTA methods and clearly showcase where their novel method outperforms other methods.

**Strengths:**

Excellent explanation of the method and figures diagramming what was done and the distinctions between training and inference periods, for both the imputation and the forecasting applications.

The algorithm's ability to be applied to previously unseen datasets/time series is a definite strength.

**Weaknesses:**

The conclusion/discussion was quite brief. I would have loved to read more about limitations and the approach for extending this to the multivariate case.

Section 3.4 was a strong inclusion of rationale for their authors' choices, but the disjoint list of conclusions and redirection to the Appendix was weak. Perhaps some (unnecessary) details of the datasets could be left to the appendix to provide space for more description of the actual method.

**Questions:**

To better compare and align this method to others in the literature, could the authors expand on statements such as what exactly they mean by how transformer models 'often suffer from significant performance degradation'? (Which performance metrics?)

Similarly, when discussing efficiency: 'less efficient than the aforementioned discrete models for regular time series' do the authors mean sample efficiency? Scalability of the algorithm to multiple dimensions or longer time series?

The 'efficient adaptation in latent space' is interesting. Is there anything to be learned about the structure of the latent spaces and how they are modified between potentially similar datasets? Which could explain perhaps why other models require a full retraining.

Would this work for imputation or forecasting not on a grid? As in, could forecasted data be easily predicted on an arbitrary grid? Is this related to the discrete performance in Table 2?

---

> ### Author Response · Authors · 2023-11-17
> **Detailed answers (1/2)**
>
> ## Weaknesses
>
>
> > Weakness 1: The conclusion/discussion was quite brief. I would have loved to read more about limitations and the approach for extending this to the multivariate case.
>
> Thank you for the question. We will provide more insight on why our model would struggle with heterogeneous multivariate time series and what can be done to overcome this challenge.
>
> - As mentioned in the conclusion, our empirical results have shown that TimeFlow is effective at handling homogeneous multivariate time series but would struggle at modeling time series with heterogeneous distributions. The key idea is that the INR learns shared parameters representing the general statistics of a family of time series and the modulation derived from the learned code $z$ adapts the INR to each sample specificity. Intuitively, shared statistics correspond to "low" frequencies, while modulations learn time series-specific "high" frequencies. If the "low" frequencies completely differ between time series (for example, let's consider a dataset where one sensor measures humidity, another sensor measures temperature, and the third sensor measures visibility), the shift modulation mechanism alone would not be sufficient to complement the statistics learned via the shared parameters.
> - One solution to overcome this challenge is to have one INR per time series distribution. All INRs can be modulated with the same vector $z$ that characterizes shared information, e.g., the location. This method is flexible but expensive if there are many different time series distributions.
>
>
> > Weakness 2: Section 3.4 was a strong inclusion of rationale for their authors' choices, but the disjoint list of conclusions and redirection to the Appendix was weak. Perhaps some (unnecessary) details of the datasets could be left to the appendix to provide space for more description of the actual method.
>
> We agree that the discussion on architecture details could be more structured, and removing some details of the dataset would leave us the necessary space. We have updated the article accordingly.
>
> ## Questions
>
>
> > Question 1: To better compare and align this method to others in the literature, could the authors expand on statements such as what exactly they mean by how transformer models 'often suffer from significant performance degradation'? (Which performance metrics?)
>
> Thank you for the question, we will make this statement clearer and provide additional experimental results to support our claim. The two sentences are important for this statement "Many state-of-the-art models, such as transformers, have been primarily designed for dense and regular grids. They struggle to handle irregular data and often suffer from significant performance degradation". This statement points out that actual SOTA methods such as Patch-TST are effective on regular grid (timesteps between observations are the same). However, if the time series are irregularly sampled or suffer from missing values, an imputation method needs to be applied to fill in the missing values and only then the forecaster can be applied. This procedure can be costly and negatively affect the performance of the forecaster.
>
> To illustrate this, we propose in Appendix D.5, a new experiment where the look-back window is sparsely observed and we forecast with Patch-TST, the strongest baseline. First, we impute the missing values in the look-back window with the linear interpolation and then we apply Patch-TST on the regular inferred grid. We evaluate this procedure against TimeFlow for different horizons and different missing values rate in the look-back window. The results in Table 21 showcase that  PatchTST performance significantly deteriorates as the sampling rate $\tau$ decreases (i.e. the number of missing values increases). In contrast, the performance of TimeFlow is only marginally affected by the reduction in the sampling rate.
>
> > Question 2: Similarly, when discussing efficiency: 'less efficient than the aforementioned discrete models for regular time series' do the authors mean sample efficiency? Scalability of the algorithm to multiple dimensions or longer time series?
>
> We meant that continuous models for time series performs worse than discrete methods on regularly sampled time series in both imputation and forecasting tasks  according to the MAE metrics on the forecast. This statement is illustrated in Tables 1 and 2 by comparing models performances. This will be made clear in the manuscript.

---

> > ### Author Response · Authors · 2023-11-17
> > **Detailed answers (2/2)**
> >
> > > Question 3: The efficient adaptation in latent space' is interesting. Is there anything to be learned about the structure of the latent spaces and how they are modified between potentially similar datasets? Which could explain perhaps why other models require a full retraining.
> >
> > Thanks for this suggestion. In order to explore and to understand the structure of the latent space, we performed a series of three new experiments with results detailed in Appendix E:
> >
> > 1. We explore how the latent space behaves between two learned codes, $z_1$ and $z_2$, which are representations of two time series $x_1$ and $x_2$. We propose in Figure 13 to visualize how new latent representations $z_{\lambda} = \lambda z_1 + (1 - \lambda) z_2$  are decoded in the time-series domain ($f_{\theta, h_w({ z_{\lambda})}}(t)$ values). We observe that the interpolation path between two codes yields a smooth transition in the time series domain. This suggests that the latent space is smooth and well-structured, and that the learned representations captured meaningful features of the time series, which could explain TimeFlow's generalization property.
> >
> > 2. We tried to interpret each dimension of the latent space independently. To do so, we perturbed a given modulation dimension (by adding Gaussian noise) only for a given layer of the INR. We then observe the difference in the time domain between the unperturbed and the perturbed time series. In Figures 14, 15, and 16, we observe three different effects on the time domain for three different perturbations of the shift modulations. Drawing conclusions for a specific code dimension for one particular INR layer is challenging.
> >
> > 3. In Appendix E.3, we visualize two time series distributions in latent space. The two distributions differ by a time shift. According to Figure 17, the larger the time shift the more the distributions are separable in the latent space. However, we note that this separability does not affect the generalization performance of TimeFlow over time, as shown by the results presented in Table 2 and Table 18.
> >
> >
> > > Question 4: Would this work for imputation or forecasting not on a grid? As in, could forecasted data be easily predicted on an arbitrary grid? Is this related to the discrete performance in Table 2?
> >
> > TimeFlow operates without needing a specific grid; it simply requires access to timestamps and corresponding values. In the experiments we used grids only for allowing comparisons between methods and also because most datasets are provided on regular grids.

---

> > > ### Comment · Reviewer_TAfq · 2023-11-23
> > >
> > > I have read the other reviewer comments and the authors responses, and am impressed with the level of detail and careful consideration of each point. I think the additional experiments and results in Appendix E are a nice addition to the paper. My score remains an 8.

---

### Official Review · Reviewer_QdLC · 2023-10-31

**Soundness:** 3 good
**Presentation:** 3 good
**Contribution:** 3 good
**Rating:** 5
**Confidence:** 4

**Summary:**

This paper proposes a new algorithm for time-series imputation and forecasting via using implicit neural representations. The proposed method particularly leverages an idea of latent modulation, extending a previous approach by making the latent vectors evolve over time. During the inference, for both imputation and forecasting, the method assumes there exist a few samples and fine-tunes the INR via auto-decoding. The paper tests the proposed methods on well-known time-series benchmark datasets and compares the result with several time-series modeling methods that can be considered as the current state-of-the-art.

**Strengths:**

- The paper is written clearly, elaborating the architecture design, and training/test algorithms.

- Time-series modeling has not been investigated much in the INR literature and this paper provides some insights that modeling time-series data in a continuous neural function can be beneficial.

- The paper compares the proposed method with several important baselines.

**Weaknesses:**

- Although the domain of the application (i.e., time-series modeling) is new and the proposed design brings an idea of the latent state evolution (in forecasting), the novelty seems to be limited. The overall architectural design follows the FFN architecture (Tancik, et al, 2020) without any consideration on how to handle multivariate time-series. Also, the idea of the latent modulation and the meta-learning-based training algorithm largely follow the previous approach (Dupont, et al, 2022). Finally, a similar idea of employing temporally-refined latent variables has been explored in (Yin, et al, 2023).

- Regarding auto-decoding process:

  - For imputation, it is natural to assume that there is a given set of measurements for a new time-series, and to set up the goal to fill-in unseen data via imputation.  For forecasting, however, the assumption of having a separate training period and a look-back window raises some concerns. Having a separate look-back window suggests that the method needs to wait until the new observations are collected to make forecasting. Some of the datasets that are considered in the paper have hourly sampling rate and this time gap might provide enough time to fine-tune other baseline models (with many model parameters, e.g., Transformers). If the ultimate goal is to achieve accurate prediction, with the given time period (an hour), fine-tuning those baselines with a new observation may provide better prediction results.

  - Similarly, another concern is fairness on the comparisons. Although it is just 3 gradient steps, auto-decoding is considered as solving an optimization problem to fine-tune the model for the new observations. What happens if the small portion of the other baselines (e.g., the last layer) is fine-tuned during the inference? For example, in forecasting, the model can be fine-tuned after making predictions on the current sliding window and then make predictions on the next sliding window with the updated models.

  - Although the method seems to provide accurate predictions both in imputation and forecasting, the method seems to struggle in predicting peaks accurately. In many applications, predicting peaks accurately would have more importance than simply minimizing MSEs (e.g., to properly prepare the electricity supply or properly set up the cost during the peak time period). Based on the eye-ball examination (Figure 5 for example), the model does not seem to provide accurate predictions in peak values.

**Questions:**

- As mentioned in the weakness section, could the author provide more justifications for performing auto-decoding while other baselines are used only for inference? Also, could the authors mention more about the fairness of the comparisons?

- Appendix section D, Tables 13 and 14 emphasize the performance degradation in Transformers models as the testing window is far apart from the training period. Would there be realistic cases where we have only the old history for training, measurements are stopped for a while, and collecting results regularly again afterwards?

- In Appendix, the paper provides experimental results for varying dimensionalities on the latent vectors and the number of gradient steps in auto-decoding during the inference. However, these experiments are performed in a limited experiment setting. Could the authors provide more insight on the effect of these hyper-parameters? For example, Table 6 provides the results with a specific dataset (Electricity) with a horizon length 720 and a look-back window length 512 and essentially says there will be no improvement after 10 or 50 gradient steps. Would this observation be valid for other datasets, for other sizes of windows? Also, what happens if the dimensionality of the latent vector is changed? Are these results also obtained by the multiple number of runs?

- Could the authors also provide justifications on not to include other time-continuous models for their baselines? Such as neural ODEs and their variants for irregular time-series modeling?

---

> ### Author Response · Authors · 2023-11-17
> **Detailed answers (1/3)**
>
> ## Weaknesses
>
> > Weakness 1: Although the domain of the application (i.e., time-series modeling) is new and the proposed design brings an idea of the latent state evolution (in forecasting), the novelty seems to be limited. The overall architectural design follows the FFN architecture (Tancik, et al, 2020) without any consideration on how to handle multivariate time-series. Also, the idea of the latent modulation and the meta-learning-based training algorithm largely follow the previous approach (Dupont, et al, 2022). Finally, a similar idea of employing temporally-refined latent variables has been explored in (Yin, et al, 2023).
>
> Thanks, we will try to highlight the differences with previous work. As indicated in the general comment, we think that our contributions involve non trivial developments and differ significantly from existing works as detailed below.
>
> 1. Comparison with [1]. In [1], the focus is on self-supervised learning: INR is only used for learning representations from static images/3D shapes that are later used for downstream tasks (classification and generation). For TimeFlow, INR is not used for learning representations but is designed for modeling contexts and dynamics in a continuous manner: this is a significantly different framework.
>
>  Concerning the specificities of  the meta-learning implementations in [1] and TimeFlow, please refer to general comment entitled "Unique Adaptation of Meta-Learning Principles".
>
>
> 2. Comparison with [2]. The frequencies embedding presented in [2] is only one of the possible INR implementations. TimeFlow uses the NERF encoding [3], which brings simplicity and good performance (we have an ablation on this topic in the paper; please see Table 4). Overall, the key is to capture multiple frequencies, and there are various alternative implementations that can achieve this. We chose one of them.
>
> 3. Comparison with [4]. In [4], authors make use of spatial INRs and the dynamics is encoded via a NeuralODE solver. Besides, the two components are trained sequentially and not end to end as we do so that the codes are fixed and not refined during training. This is essentially different from TimeFlow were for each new dynamics, the latent code is updated according to the look-back window to capture the local dynamics.
>
> [1] E. Dupont, et al. From data to functa: Your data point is a function and you can treat it like one. ICML 2022.
>
> [2] Tancik, Matthew, et al. Fourier features let networks learn high frequency functions in low dimensional domains. NeurIPS 2020.
>
> [3] Mildenhall, Ben, et al. Nerf: Representing scenes as neural radiance fields for view synthesis. Communications of the ACM.
>
> [4] Y. Yin, et al. Continuous pde dynamics forecasting with implicit neural representations. ICLR 2023.
>
>
>
> > Weakness 2: For forecasting, however, the assumption of having a separate training period and a look-back window raises some concerns. Having a separate look-back window suggests that the method needs to wait until the new observations are collected to make forecasting. Some of the datasets that are considered in the paper have hourly sampling rate and this time gap might provide enough time to fine-tune other baseline models (with many model parameters, e.g., Transformers). If the ultimate goal is to achieve accurate prediction, with the given time period (an hour), fine-tuning those baselines with a new observation may provide better prediction results.
>
> If we understand correctly this question, concerns arise from Tables 2 and 18, where we train over a given period and then test over multiple periods in the future. Indeed we make the assumption that the models will not be retrained during a given operational period, which corresponds we believe to most real world applications. This is a classical forecasting framework. Then in any case, forecasting is performed at a given horizon, from an adjacent look-back window. This holds  for any forecasting model.
> Please tell us if we misunderstood your remark.

---

> ### Author Response · Authors · 2023-11-17
> **Detailed answers (2/3)**
>
> > Weakness 3: Similarly, another concern is fairness on the comparisons. Although it is just 3 gradient steps, auto-decoding is considered as solving an optimization problem to fine-tune the model for the new observations. What happens if the small portion of the other baselines (e.g., the last layer) is fine-tuned during the inference? For example, in forecasting, the model can be fine-tuned after making predictions on the current sliding window and then make predictions on the next sliding window with the updated models. + (Q.2) As mentioned in the weakness section, could the author provide more justifications for performing auto-decoding while other baselines are used only for inference? Also, could the authors mention more about the fairness of the comparisons?
>
> Auto-decoding does not perform any fine tuning and all the models including baselines are evaluated in the same conditions. In a forecasting setting, the objective is to forecast future values (horizon) conditioned on past ones (look-back window). A forecaster could make use of any encoding of the look-back information. For example one could use the look-back window values as such with no change, or one can encode the look-back values in a compact representation used as input for a forecaster (a simple compact representation would be for example a sum of the look-back values). Auto-decoding is just another way of computing this encoding. In order to better detail why auto-encoding does not perform any form of fine tuning, let us now try an alternative rewriting of the auto-decoding that might make things clearer. Let $t^{L}$ stands for the look-back window timestamps and $t^{H}$ stands for the horizon timestamps. We consider for sake of simplicity only one gradient step to encode $z$ according the look-back window values $x^{L}$. The TimeFlow forecaster can be written as a deterministic function of the look-back window:
> $$
>     f(t^{H}; \theta, w, z(x^{L})) \; \text{where} \; z(x^{L}) = z^{(0)} - \alpha \nabla_{z}\Vert f_{\theta, h_{w}(z^{(0)})}(t^{L}) - x^{L}\Vert^{2}_{2}
> $$
> Making 1, 3, or 10 adaptation gradient steps uniquely changes the complexity of the function encoding the code $z$.
>
> As a side note, DeepTime also uses this type of encoding. Time-step-based models that don't have a structure with an explicit set encoder need to pass the look-back window information somehow.
>
> > Weakness 4: Although the method seems to provide accurate predictions both in imputation and forecasting, the method seems to struggle in predicting peaks accurately. In many applications, predicting peaks accurately would have more importance than simply minimizing MSEs (e.g., to properly prepare the electricity supply or properly set up the cost during the peak time period). Based on the eye-ball examination (Figure 5 for example), the model does not seem to provide accurate predictions in peak values.
>
> You are right; in many real-world applications, accurate peak prediction would be more important than minimizing MSEs.
>
> We believe that this kind of "smooth" prediction is more a matter of the choice of the optimized loss (in our case, MSE) than the model. For example, in Figure 10, where we compare the quality of the TimeFlow and PatchTST predictions, both methods have trouble catching the peaks.
>
> If the observations are regularly spaced, one way to overcome this challenge is to train the models with more appropriate but more expensive losses, such as Soft-DTW [1] or DILATE [2], which are differentiable. If the observations are not regularly spaced, to the best of our knowledge, finding a differentiable loss that can penalize shape pattern error is an open problem in time series.
>
> [1] M. Cuturi, et al. Soft-dtw: a differentiable loss function for time-series. ICML 2017.
>
> [2] V. Le Guen, et al. Shape and time distortion loss for training deep time series forecasting models. Neurips 2019.

---

> ### Author Response · Authors · 2023-11-17
> **Detailed answers (3/3)**
>
> ## Questions
>
> > Question 1: As mentioned in the weakness section, could the author provide more justifications for performing auto-decoding while other baselines are used only for inference? Also, could the authors mention more about the fairness of the comparisons?
>
> Please see answer to Weakness 3.
>
> > Question 2: Appendix section D, Tables 13 and 14 emphasize the performance degradation in Transformers models as the testing window is far apart from the training period. Would there be realistic cases where we have only the old history for training, measurements are stopped for a while, and collecting results regularly again afterwards?
>
> Please see answer to Weakness 2.
>
> > Question 3: In Appendix, the paper provides experimental results for varying dimensionalities on the latent vectors and the number of gradient steps in auto-decoding during the inference. However, these experiments are performed in a limited experiment setting. Could the authors provide more insight on the effect of these hyper-parameters? For example, Table 6 provides the results with a specific dataset (Electricity) with a horizon length 720 and a look-back window length 512 and essentially says there will be no improvement after 10 or 50 gradient steps. Would this observation be valid for other datasets, for other sizes of windows? Also, what happens if the dimensionality of the latent vector is changed? Are these results also obtained by the multiple number of runs?
>
> Thanks for the question, you are right. It is important to perform these ablations across all datasets and horizons, we have then added several new experiments as detailed below:
>
> - Regarding the ablation on the $z$ dimension. In Appendix A.2.2 we added Table 5, which investigates the impact of $z$ dimensionality on the forecasting performance of TimeFlow. We performed experiments on all the datasets for each horizon with different $z$ dimensions (32, 64, 128, 256). The results suggest that a $z$ dimension of 128 is a reasonable compromise but only optimal for some settings. Moreover, even though the choice of $z$ dimension seems important, it doesn't critically impact the MAE error for the forecasting task.
> - Regarding the number of gradient steps within the inner loop. In Appendix A.2.3 we have added Tables 7, 8, and 9. In Table 7, TimeFlow is trained with a single gradient step in the inner loop and tested with various gradient steps (1, 3, 10, 50) in the inner loop at inference. We evaluate this over all datasets and horizons. Tables 8 and 9 show the same results, where the number of gradient steps in the inner loop during training is 3 and 10, respectively.
> Two conclusions can be drawn from these results:
>     - Keeping the number of gradient steps the same for training and inference is a good option.
>     - Three gradient steps in the inner loop seems to be a good choice regarding MAE results while being faster than ten gradient steps.
>
> > Question 4: Could the authors also provide justifications on not to include other time-continuous models for their baselines? Such as neural ODEs and their variants for irregular time-series modeling?
>
> Neural ODEs and their variants for irregular time-series modeling have been shown to be outperformed by CSDI [1] and mTAN [2] for the imputation task. CSDI and mTAN are, respectively, discrete and continuous methods and been re-implemented in our work and are used as baselines. In addition, we would like to highlight that we have re-implemented 11 recent deep learning baselines in this paper.
>
> [1]: Tashiro, Yusuke, et al. CSDI: Conditional score-based diffusion models for probabilistic time series imputation. NeurIPS 2021.
>
> [2]: Shukla, Satya Narayan, and Benjamin M. Marlin. "Multi-time attention networks for irregularly sampled time series." ICLR 2021.

---

> > ### Comment · Reviewer_QdLC · 2023-11-23
> >
> > I'd like to thank the authors for providing their responses. After reading the responses and other comments by the reviewers, I decided to keep the current rating.
> >
> > While not intending to introduce another discussion phase, I'd like to clarify my question regarding "fine-tuning". The proposed model performs auto-decoding in the inference phase, which requires the number of optimization steps, searching for a latent representation that minimizes the objective. Through this process, the latent vector is being updated. So as the authors said, it is valid to say that there is no model parameters being updated. But at the same time, updating a latent vector in the latent modulation scheme is equivalent to having a different realization of an INR, where the different model parameters of the INR obtained through modulation, and this can be interpreted as "fine-tuning".
> >
> > In other models, the trained models are being used to make predictions without any further update on the model parameters (i.e., making a prediction is equivalent to taking a forward pass and no additional optimization process during the inference phase.) I was asking what happens if the trained models (or some parts of them, e.g., the last layer) can be updated (or fine-tuned) with the same (similar) amount of the computational with the given additional data points in the inference phase.
> >
> > Again, thank you for the authors' detailed response.

---

> ### Author Response · Authors · 2023-11-23
> **Additional response**
>
> Thank you for your response, in a final attempt to convince you of the fairness of our setting, we would like to clarify three points.
>
>
> First, in our experimental setting there is no available data to perform fine-tuning for any of the models used in the comparison. Indeed, this setup would assume that we have some pairs of look-back / horizon time series values $(x_L, x_H)_j$ for each temporal window to update the model parameters. This is not the case in our study: at inference we have only a new look back window that is observed and we must forecast with this information only. Therefore, all models behave similarly at inference and encode or auto-decode (DeepTime and TimeFlow) the information contained in the look-back window.
>
>
> Second, modulating or adapting the INR weights is a popular way to handle multiple signals with the same coordinate-based network and does not perform finetuning. For fairness, the Neural Process (NP) architecture that we used employs the same FFN as TimeFlow, i.e. with the shift modulations, the only difference being that NP uses a set-encoder instead to find the $z$. Still, as you can see, the latent code $z$ also adapts the weights of the network, otherwise it would not be able to represent a time series on a new look-back window. This is also the case with DeepTime, which by definition has no projection layer $W$. This projection layer $W$ needs to be found by solving a least-square problem given a time series observed on the look-back window $x_L$. Again, in this case, the coordinate-based network needs to adapt some of its weights to infer the values given the look-back window.
>
> Third, we could not perform fine tuning as you asked because it was not aligned with our experimental setting, i.e. no available data. However, we would like to point out that we performed forecasting experiments where the horizon of the test period was located just after the training period while the look-back window is actually in the training period. In other words, in this setting there would be no need to fine tune the model as they are already trained with all the data available. The results can be found in Table 17, but we provide them below for readability.
>
> As you can see, in this setting TimeFlow is the best model along with PatchTST. AutoFormer and Informer behave much better than in Table 2 but are still outperformed by PatchTST, DLinear and TimeFlow. If you consider this setting to be the most representative in terms of forecasting performance, we can put Table 17 inside the core manuscript and switch its place with Table 2.

---

> > ### Author Response · Authors · 2023-11-23
> > **Table 17**
> >
> > |                     |            | Continuous methods |                   |                   | Discrete methods   |                   |                   |                   |
> > |---------------------|------------|--------------------:|-------------------:|-------------------:|--------------------:|-------------------:|-------------------:|-------------------:|
> > |                     | $H$        | TimeFlow           | DeepTime          | Neural Process    | Patch-TST           | DLinear           | AutoFormer        | Informer           |
> > |---------------------|------------|--------------------:|-------------------:|-------------------:|--------------------:|-------------------:|-------------------:|-------------------:|
> > | **Electricity**     | 96         | 0.218 $\pm$ 0.017  | 0.240 $\pm$ 0.027 | 0.392 $\pm$ 0.045 | **0.214 $\pm$ 0.020** | 0.236 $\pm$ 0.035 | 0.310 $\pm$ 0.031 | 0.293 $\pm$ 0.0184 |
> > |                     | 192        | 0.238 $\pm$ 0.012  | 0.251 $\pm$ 0.023 | 0.401 $\pm$ 0.046 | **0.225 $\pm$ 0.017** | 0.248 $\pm$ 0.032 | 0.322 $\pm$ 0.046 | 0.336 $\pm$ 0.032  |
> > |                     | 336        | 0.265 $\pm$ 0.036  | 0.290 $\pm$ 0.034 | 0.434 $\pm$ 0.075 | **0.242 $\pm$ 0.024** | 0.284 $\pm$ 0.043 | 0.330 $\pm$ 0.019 | 0.405 $\pm$ 0.044  |
> > |                     | 720        | 0.318 $\pm$ 0.073  | 0.356 $\pm$ 0.060 | 0.605 $\pm$ 0.149 | **0.291 $\pm$ 0.040** | 0.370 $\pm$ 0.086 | 0.456 $\pm$ 0.052 | 0.489 $\pm$ 0.072  |
> > | **SolarH**          | 96         | **0.172 $\pm$ 0.017** | 0.197 $\pm$ 0.002 | 0.221 $\pm$ 0.048 | 0.232 $\pm$ 0.008 | 0.204 $\pm$ 0.002 | 0.261 $\pm$ 0.053 | 0.273 $\pm$ 0.023  |
> > |                     | 192        | **0.198 $\pm$ 0.010** | 0.202 $\pm$ 0.014 | 0.244 $\pm$ 0.048 | 0.231 $\pm$ 0.027 | 0.211 $\pm$ 0.012 | 0.312 $\pm$ 0.085 | 0.256 $\pm$ 0.026  |
> > |                     | 336        | 0.207 $\pm$ 0.019  | **0.200 $\pm$ 0.012** | 0.241 $\pm$ 0.005 | 0.254 $\pm$ 0.048 | 0.212 $\pm$ 0.019 | 0.341 $\pm$ 0.107 | 0.287 $\pm$ 0.006  |
> > |                     | 720        | **0.215 $\pm$ 0.016** | 0.240 $\pm$ 0.011 | 0.403 $\pm$ 0.147 | 0.271 $\pm$ 0.036 | 0.246 $\pm$ 0.015 | 0.368 $\pm$ 0.006 | 0.341 $\pm$ 0.049  |
> > | **Traffic**         | 96         | 0.216 $\pm$ 0.033  | 0.229 $\pm$ 0.032 | 0.283 $\pm$ 0.028 | **0.201 $\pm$ 0.031** | 0.225 $\pm$ 0.034 | 0.299 $\pm$ 0.080 | 0.324 $\pm$ 0.113  |
> > |                     | 192        | 0.208 $\pm$ 0.021  | 0.220 $\pm$ 0.020 | 0.292 $\pm$ 0.023 | **0.195 $\pm$ 0.024** | 0.215 $\pm$ 0.022 | 0.320 $\pm$ 0.036 | 0.321 $\pm$ 0.052  |
> > |                     | 336        | 0.237 $\pm$ 0.040  | 0.247 $\pm$ 0.033 | 0.305 $\pm$ 0.039 | **0.220 $\pm$ 0.036** | 0.244 $\pm$ 0.035 | 0.450 $\pm$ 0.127 | 0.394 $\pm$ 0.066  |
> > |                     | 720        | **0.266 $\pm$ 0.048** | 0.290 $\pm$ 0.045 | 0.339 $\pm$ 0.037 | **0.268 $\pm$ 0.050** | 0.290 $\pm$ 0.047 | 0.630 $\pm$ 0.043 | 0.441 $\pm$ 0.055  |
> > | **TimeFlow improvement** |            | /                  | 6.56 $\%$          | 30.79 $\%$         | 2.64 $\%$           | 7.30 $\%$          | 35.43 $\%$         | 33.07 $\%$         |

---

### Official Review · Reviewer_FJUP · 2023-10-31

**Soundness:** 4 excellent
**Presentation:** 3 good
**Contribution:** 2 fair
**Rating:** 5
**Confidence:** 4

**Summary:**

The problem setting is time series data, with irregular sampling and missing data. The paper proposes a method that learns a (conditional) implicit neural representation for time series. The model can be used for forecasting and imputation. It shows promising results on these tasks, compared to other baselines.

**Strengths:**

Since the method is a pretty straightforward application of Dupont et al. (2022) to time series, the approach is sound.

The results on imputation are decent. The methods shows promising results in terms of MAE and the imputation in Figure 3 looks good compared to BRITS. However, it seems there is still a lot of performance improvement left on the table. Electricity is a pretty simple periodic dataset so I can imagine achieving better results with further tuning or some other model.

Forecasting results are again good but the model is only matching the competitors. Forecasting + imputation is showing even better results and has some potential real-world applications. However, some other models can be included in the comparison here.

**Weaknesses:**

The approach has limited novelty since it's mostly building upon known previous work. This same architecture can be applied to images, point clouds, and so on. Although the discussion of implementation choices is a nice addition, they are again not necessarily time series dependent.

Results on imputation are decent, but the method is not beating other baselines most of the time. It is usually close to BRITS and some other baselines. This might indicate used datasets are too simple. Also, using such regular data for imputation is not ideal since one of the points of the proposed method is that it can handle irregular sampling rate. Something like MIMIC dataset might be a better choice, especially since it already contains missing values.

According to Table 16, the model is more costly compared to already costly transformer-based models.

According to Table 17, PatchTST is often outperforming proposed method which means that it's better at adapting to new time series, contrary to what is stated in the main text.

As a side note, it would be interesting to have a probabilistic version of this model.

The biggest drawbacks of this paper are lack of novelty, not so stellar results and not applying this method to actual continuous-in-time data.

**Questions:**

- Can you compute MAE in Figure 3 for a naive baseline that simply connects training points with a line?

- Can you explain why the results in Table 13 and 14 differ for AutoFromer and Informer?

- If I understand the setting in 4.3 correctly, all models for imputation from 4.1 should be able to produce imputed values and forecast. Then, they should be included in Table 2.

---

> ### Author Response · Authors · 2023-11-17
> **Detailed answers (1/2)**
>
> ## Answers to comments in the strengths section
>
> > Strength 0: Since the method is a pretty straightforward application of Dupont et al. (2022) to time series, the approach is sound.
>
> As stated in the general answer, we advocate that our work is not a direct adaptation of [1]. Let us detail more precisely how it differs from [1].
>
> 1. Their focus is on self-supervised learning: INR is only used for learning representations from images/3D shapes that are later used for downstream tasks with the focus on classification and generation. Besides they only consider static data. For TimeFlow, INR is not used for learning representations but it is designed for modeling contexts and dynamics in a continuous manner: this is a significantly different framework.
>
> 2.  Concerning the specificities of  the meta-learning implementations in [1] and TimeFlow, please refer to general comment entitled "Unique Adaptation of Meta-Learning Principles".
>
> [1]  E. Dupont, et al. From data to functa: Your data point is a function and you can treat it like one. ICML 2022.
>
> > Strength 1: The results on imputation are decent. The methods shows promising results in terms of MAE and the imputation in Figure 3 looks good compared to BRITS. However, it seems there is still a lot of performance improvement left on the table. Electricity is a pretty simple periodic dataset so I can imagine achieving better results with further tuning or some other model.
>
>  We compare TimeFlow with 7 state-of-the-art methods. We beat these methods 13 times out of 15 and improved by 19% the imputation score relatively to the best baseline (BRITS).
>
>  To quantify the complexity of the imputation settings for the considered datasets, we present Table 16 in Appendix C.4. This table compares TimeFlow, Linear Interpolation, and KNN Interpolation for all imputation settings. Linear interpolation is used to quantify the variability within the observed time series, while KNN interpolation assesses the similarity between these time series. These basic methods effectively highlight the complexity of the imputation task. This comparison emphasizes TimeFlow performances for low sampling rates.
>
> > Strength 2: Forecasting results are again good but the model is only matching the competitors. Forecasting + imputation is showing even better results and has some potential real-world applications. However, some other models can be included in the comparison here.
>
> We compare TimeFlow to 6 state-of-the-art methods for the forecasting task with dense regular look-back windows. Our model outperforms 5 out of 6 and is on par with Patch-TST. As a conclusion, TimeFlow is as good as the SOTA discrete method  Patch-TST on regularly sampled time series, while being able to handle irregular situations where Patch-TST fails.
>
>
>
> ## Weaknesses
>
> > Weakness 1: The approach has limited novelty since it's mostly building upon known previous work. This same architecture can be applied to images, point clouds, and so on. Although the discussion of implementation choices is a nice addition, they are again not necessarily time series dependent.
>
> As already detailed we respectfully disagree with this argument. As detailed in the general comment and in response to strength 0, while the individual blocks of TimeFlow may be already known, their use and combination for time series involve original developments. Notably, the INR algorithms designed for images, point clouds, and similar domains are not directly applicable to TimeFlow due to their lack of consideration for temporal dynamics.
>
> > Weakness 2: Results on imputation are decent, but the method is not beating other baselines most of the time. It is usually close to BRITS and some other baselines. This might indicate used datasets are too simple. Forecasting results are again good but the model is only matching the competitors. Also, using such regular data for imputation is not ideal since one of the points of the proposed method is that it can handle irregular sampling rate. Something like MIMIC dataset might be a better choice, especially since it already contains missing values.
>
> We are surprised by this argument. Across the three datasets under consideration, our proposed method outperforms the 7 SOTA baselines in 13 out of 15 settings. The average improvement over the best baseline, BRITS, amounts to approximately 19%.

---

> ### Author Response · Authors · 2023-11-17
> **Detailed answers (2/2)**
>
> > Weakness 3: According to Table 16, the model is more costly compared to already costly transformer-based models.
>
> Table 20 (formerly 16) shows the inference results. Indeed, due to code adaptation, TimeFlow takes longer at inference than transformers. However, as specified in section A.2.3, TimeFlow can infer $321$ (samples) $* (720 + 512)$ timestamps, which represents around 400k values in less than 0.2 seconds on a single GPU (NVIDIA TITAN RTX GPU 24Go). This performance looks acceptable for real-world applications.
>
> On the memory consumption side, as shown in Table 19, TimeFlow has half the weight of Informer and Autoformer and 2 to 10 times less than PatchTST.
>
> > Weakness 4: According to Table 17, PatchTST is often outperforming proposed method which means that it's better at adapting to new time series, contrary to what is stated in the main text.
>
> The relative mean improvement of TimeFlow w.r.t. PatchTST in Table 21, is less than -1%. We conclude that TimeFlow and PatchTST perform equally well in this setting. Note that these experiments  consider regularly time sampled data. To conclude TimeFlow does as well as PatchTST for the regular setting with new time series while the latter cannot handle irregular or continuous settings as TimeFlow does.
>
> > Weakness 5: As a side note, it would be interesting to have a probabilistic version of this model.
>
> We agree, but this is beyond the scope of this paper.
>
> > Weakness 6: The biggest drawbacks of this paper are lack of novelty, not so stellar results and not applying this method to actual continuous-in-time data.
>
> Please see previous answers.
>
> ## Questions
>
> > Question 1: Can you compute MAE in Figure 3 for a naive baseline that simply connects training points with a line?
>
> Thanks for the suggestion. We included a linear interpolation as well as a KNN baseline for the imputation task in Table 16. For the Electricity dataset, at a 10% sampling rate (10% of observed values in the grid), TimeFlow score is 0.250 $\pm$ 0.010, BRITS score is 0.287 $\pm$ 0.015 and linear interpolation score is 0.716 $\pm$ 0.039. More specifically, in Figure 3, the linear interpolation MAE for top figure is 0.801 and 0.767 for the bottom figure.
>
> > Question 2: Can you explain why the results in Table 13 and 14 differ for AutoFormer and Informer?
>
> Table 17 (formerly Table 13) showcases results for the setting where the test window is adjacent to the training period. For Figure 7, this means that the test window is only the test period n°1. Table 18 (formerly Table 14) showcases results where the test windows are disjoint from the training period. For Figure 7, this means that the test windows are test periods n°2, n°3, n°4 and n°5. The results for each model may differ between these two tables because the test periods are different. However, we can see that while the other methods experience a slight increase in their MAE score, Informer and AutoFormer experience a significant deterioration in their forecasting performance. These methods seem to generalize poorly when tested on new time periods.
>
> > Question 3: If I understand the setting in 4.3 correctly, all models for imputation from 4.1 should be able to produce imputed values and forecast. Then, they should be included in Table 2.
>
>  The models in section 4.1 are designed for imputation only and cannot forecast. Table 2 concerns only forecasting methods, and section 4.3 concerns continuous methods evaluated in a setting where they must jointly impute missing values from the look-back window and forecast over the horizon.

---

> > ### Comment · Reviewer_FJUP · 2023-11-22
> >
> > Thank you for your detailed answers and including the additional experiments.
> >
> > I still think that limited novelty and lack of actual continuous time experiments remain as the main issues.
> >
> > One of the minor issues is the difference between forecasting and imputation. In my view you can forecast by imputing "to the right" of your observed data. This is what you use in 4.3 and use "pure imputation" baseline (NP) for this. So I would say that you should include any imputation model here. But for an even better assessment of your forecasting abilities you should include pure forecasting models that work with missing historical values. And again, the novelty is overstated when you say this is "first continuous framework ... that successfully integrates imputation and forecasting".
> >
> > I would not mind seeing this paper accepted in some form, with less emphasis on forecasting and more on imputation but for now I will keep my score.

---

> > > ### Author Response · Authors · 2023-11-22
> > > **Answer**
> > >
> > > Thank you for your response. We are pleased to know that our additional experiments and answers have provided further insights.
> > >
> > > > One of the minor issues is the difference between forecasting and imputation.
> > >
> > > Regarding the selection of baselines in Tables 1 and 2, we acknowledge your perspective. Our approach was to consider baselines that specialize in either imputation or forecasting. We understand your point that forecasting could be seen as a specific type of imputation task, particularly when no observations are available "to the right." In each case, our aim was to identify the most pertinent baselines for long-term forecasting and imputation.
> > > In Table 3, our objective was to demonstrate that forecasters trained specifically for forecasting struggle when their look-back window includes missing values. Here, we utilized a Neural Process (NP) trained on forecasting tasks. We initially did not include pure imputation baselines in Table 3, as we believed this might lead to an unfair comparison in a forecasting context. However, your suggestion is insightful.
> > > For the final version of the manuscript, we plan to adapt the imputation baselines for the forecasting task. This will allow for a more comprehensive comparison in Table 3.
> > >
> > > >  the novelty is overstated when you say this is "first continuous framework ... that successfully integrates imputation and forecasting.
> > >
> > > Indeed, we can clarify this point in the final version of the manuscript. We won't claim that our model is the first continuous model for joint imputation and forecasting. We will also put more emphasis on the adaptation of INR for efficient continuous modeling of time series.
> > >
> > >
> > > We appreciate your valuable feedback and are committed to refining our manuscript accordingly.

---

### Author Response · Authors · 2023-11-17
**General comment**

We thank all the reviewers for their comments and suggestions. We carefully answered all the questions. We have also added new experimental results and modified the manuscript (changes are highlighted with a different color) following the suggestions of the reviewers.

## Changes to the manuscript

- reviewer FJUP's suggestions.
    - Additional imputation baselines to benchmark the complexity of the imputation problem (Table 16; Appendix C.4).
    - More representative score for TimeFlow improvement (Appendix B).
- reviewer QdLC's suggestions.
    - More ablations on the number of inner-steps across all datasets (Tables 7, 8, and 9; Appendix A.2.3).
    - More ablations on the influence of the latent space dimension (Table 5; Appendix A.2.2)
- reviewer TAfq's suggestions.
    - Latent space interpretation through various qualitative results (Appendix E) .
    - Inability of discrete forecasting baselines to handle missing values in the look-back window (Table 22; Appendix D.5).
    - More details in implementation choices (Section 3.4).

Overall we would like to address the concerns about the novelty of our approach.

## Novelty of TimeFlow

**Innovative Application of Meta-Learning in Time Series Analysis with TimeFlow:**

Reviewers FJUP and QdLC question the novelty of the contribution. While the individual blocks are already known, their use and combination for time series involve original developments. While it draws upon the established concepts of modulated Implicit Neural Representations (INRs) and meta-learning, TimeFlow is far from a trivial adaptation of existing methods. We highlight in the following the novelty of the proposed approach.

1. **Distinct Focus and Application**:
TimeFlow employs INRs to continuously model the dynamics and context of time series data. This approach marks a significant departure from the traditional use of INRs. [1] for example, primarily uses INRs for self-supervised learning tasks in static data contexts that are then used for downstream tasks such as image or 3D shape classification and generation.

2. **Unique Adaptation of Meta-Learning Principles**: TimeFlow is indeed grounded in the general meta-learning principles introduced in the seminal MAML paper [2]. The latter has been used and adapted to various situations in numerous publications. [1] and TimeFlow are then different extensions of MAML. Meta-learning performs a two stage optimization: inner loop and outer loop perform optimization on sample sets from the same distributions. In TimeFlow, the inner loop operates on conditioning data (lookback window) while the outer loop does it on objective data (horizon window). The distributions in the loops are then different in our case due to the temporal shift. MAML cannot handle this case, while our adaptation is specifically defined for it.

3. **Advancement for Time Series Modeling**: TimeFlow stands out as the first continuous framework in the time series domain that successfully integrates imputation and forecasting within a single, unified model, effectively handling irregular time series through continuous modeling. This is also the first continuous model able to compete with SOTA discrete models on regular while being able to handle situation where these models fail.

[1]  E. Dupont, et al. From data to functa: Your data point is a function and you can treat it like one. ICML 2022.

[2] Finn, et al. Model-agnostic meta-learning for fast adaptation of deep networks. International conference on machine learning. PMLR, 2017.

---

### Author Response · Authors · 2023-11-21
**Discussion period ending soon**

Dear reviewers,

As we approach the conclusion of the discussion period, we wish to emphasize that we have diligently addressed all of your comments. This has been achieved through the inclusion of new experiments and substantial updates to our manuscript. We are willing to respond to any additional questions or feedback you may have.
Thank you for your insights.

Warm regards.

---

### Meta-Review · Area_Chair_WCSa · 2023-12-06

**Metareview:**

The paper proposes a new algorithm for time-series imputation and forecasting via using implicit neural representations. The proposed method particularly leverages an idea of latent modulation, extending a previous approach by making the latent vectors evolve over time. The authors answered some of the questions of the reviewers. However, the reviewers still think that the design and the training algorithm very much inherit latent-modulation and the meta-learning algorithm of Functa (Dupont, et al, 2022, ICML) and the inference via auto-decoding is the same. Along with the novelty, auto-decoding with the training data outside of the training region during the inference period is still concerning. While being able to perform auto-decoding can be considered as benefits of using INR + latent modulation, at the same time, it uses the data in the test region to solve an optimization problem to perform inference, which brings some questions what happens if the other models utilize the same amount of data to fine-tune (solving an optimization problem) to update a portion of model parameters. Most of the reviewers do not support the paper and still think that the paper is not sufficient to be published at the ICLR. Based on the reviewers' suggestions, I cannot recommend acceptance of the paper but I encourage the authors to resubmit it to the next venues.

**Justification For Why Not Higher Score:**

Most of the reviewers do not support the paper.

**Justification For Why Not Lower Score:**

N/A

---

### Decision · Program_Chairs · 2024-01-16

Reject